# CarbonGlobe: A Global-Scale, Multi-Decade Dataset and Benchmark for Carbon Forecasting in Forest Ecosystems

**Zhihao Wang**[1], **Lei Ma**[1], **George Hurtt**[1], **Xiaowei Jia**[2], **Yanhua Li**[3]
**Ruohan Li**[1], **Zhili Li**[1], **Shuo Xu**[1], **Yiqun Xie**[1*]
[1]University of Maryland, [2]University of Pittsburgh, [3]Worcester Polytechnic Institute
{zhwang1, lma6, gchurtt, r526li, lizhili, shuoxu98, xie}@umd.edu,
xiaowei@pitt.edu, yli15@wpi.edu

## Abstract

Forest ecosystems play a critical role in the Earth system as major carbon sinks that are essential for carbon neutralization and climate change mitigation. However, the Earth has undergone significant deforestation and forest degradation, and the remaining forested areas are also facing increasing pressures from socioeconomic factors and climate change, potentially pushing them towards tipping points. Responding to the grand challenge, a theory-based Ecosystem Demography (ED) model has been continuously developed over the past two decades and serves as a key component in major initiatives, including the Global Carbon Budget, NASA Carbon Monitoring System, and US Greenhouse Gas Center. Despite its growing importance in combating climate change and shaping carbon policies, ED's expensive computation significantly limits its ability to estimate carbon dynamics at the global scale with high spatial resolution. Recently, machine learning (ML) models have shown promising potential in approximating theory-based models with interesting success in various domains including weather forecasting, thanks to the open-source benchmark datasets made available. However, there are currently no publicly available ML-ready datasets for global carbon dynamics forecasting in forest ecosystems. The limited data availability hinders the development of corresponding ML emulators. Furthermore, the inputs needed for running ED are highly complex with over a hundred variables from various remote sensing products. To bridge the gap, we develop a new ML-ready benchmark dataset, *CarbonGlobe*, for carbon dynamics forecasting, featuring that: (1) the data has a global-scale coverage at $0.5°$ resolution; (2) the temporal range spans 40 years; (3) the inputs integrate extensive multi-source data from different sensing products, with calibrated outputs from ED; (4) the data is formatted in ML-ready forms and split into different evaluation scenarios based on climate conditions, etc.; (5) a set of problem-driven metrics is designed to develop benchmarks using various ML models to best align with the needs of downstream applications. Our dataset and code are publicly available on Kaggle and GitHub: https://www.kaggle.com/datasets/zhihaow/carbonglobe and https://github.com/zhwang0/carbon-globe.

## 1 Introduction

Forest ecosystems play critical roles in the Earth system by offering a wide array of essential ecosystem services such as carbon storage and climate change mitigation [45, 37]. Studies have

---

*Corresponding author.

39th Conference on Neural Information Processing Systems (NeurIPS 2025) Track on Datasets and Benchmarks.

shown that terrestrial ecosystems, including forests, sequestered about one-third of global fossil fuel emissions in the past decades [15]. However, the Earth has undergone significant deforestation and forest degradation as a result of human activities and climate change [7, 60]. The remaining forested areas are also facing increasing pressures from socioeconomic factors and climate change, and could be pushed to tipping points [24, 20]. Thus, there is a growing interest in maintaining forest carbon sequestration through afforestation and reforestation initiatives from local to global scales [12]. Climate mitigation policies now also include significant international commitments to afforestation, reforestation, and improved management practices, including the "one trillion trees" project [3], which are necessary to achieve the net zero emission goal. These developments have led to a strong demand for advancing the understanding of forest carbon dynamics and its capacity to mitigate future climate change.

The Ecosystem Demography (ED) v3.0 model is a pioneering model based on ecological and carbon cycle theories, and it has been continually developed over the past decades to improve terrestrial carbon dynamic modeling [25, 14, 38]. ED uniquely takes in observations from remote sensing satellites and in-situ sensors (e.g., meteorological conditions, initial tree heights, soil properties) to forecast essential carbon-related variables such as carbon stocks, carbon fluxes, and carbon sequestration potential. ED has been extensively calibrated and evaluated against global and regional real observations, including vegetation structure, carbon fluxes, LiDAR data, and forest inventories [22, 39, 38, 16]. Due to its high quality, ED has been used to support NASA's Carbon Monitoring System and included in the official Global Carbon Budget [16, 17]. ED results are also operationally adopted by the State of Maryland, US, for annual forest carbon inventory updates [2, 1, 23].

Despite its growing importance in combating climate change and shaping carbon policies, ED's expensive computation has become a bottleneck constraining its capability to estimate and forecast carbon dynamics globally at high spatial resolutions. This capability, however, is important as policy-making and management practices often require fine-grained outputs, which are unavailable from coarse resolution maps due to spatial heterogeneity [22, 40]. This is a missed opportunity as a unique characteristic of ED is its ability to model complex plant-scale carbon dynamics and integrate them into large-scale climate submodules such as the carbon cycle. Recently, machine learning (ML) models (e.g., FourCastNet [31], GraphCast [33]) have shown promising potential in approximating theory-based models with interesting success in various domains including weather forecasting, thanks to the open-source benchmark datasets made available, such as the ERA5 dataset from WeatherBench 2 [47]. However, there are yet any publicly available ML-ready datasets for the critical application of carbon dynamics forecasting in forest ecosystems, where we consider a **"ML-ready dataset"** as one that has gone through preprocessing by domain and AI experts and converted into a ready-to-use input-output format for ML models. In addition, the architectures of deep learning emulators for theory-based models often need targeted designs to capture the characteristics of the physical processes (e.g., Fourier neural operators for long-distance convolution [31]). The limited availability of ML-ready data also hinders the development of corresponding network architectures. Furthermore, while ED and its inputs are publicly available, it is a highly complex task to collect, process, and integrate all $100^+$ input variables at large scales and over long periods from different sources and types of data products, including meteorological conditions, 3D height profiles, etc.

This paper introduces *CarbonGlobe*, the first global-scale ML-ready dataset for carbon dynamics estimation and forecasting in forest ecosystems. *CarbonGlobe* integrates a comprehensive set of variables covering different aspects of the physiological and ecological processes involved, including meteorological variables from NASA Daymet and MERRA2, soil properties from the POLARIS dataset, $CO_2$ concentration from NOAA CarbonTracker, and many more (details in Sec. A.3). These diverse inputs are used to generate and calibrate the outputs of ED in high-performance-computing (HPC) environments. The complete *CarbonGlobe* spans an extensive period of 40 years under diverse initial forest conditions. The ML-ready dataset is prepared together by carbon and ML experts to make it easily accessible to ML researchers, while ensuring the data quality aligned with domain science and applications. We also design problem-driven metrics and implement a suite of ML forecasting models to develop extensive benchmarks with different scenarios for evaluation and comparison. The open-source project serves as a foundation for future model developments. The dataset and code are publicly available on Kaggle `https://www.kaggle.com/datasets/zhihaow/carbonglobe` and GitHub `https://github.com/zhwang0/carbon-globe`. Our contributions are summarized as follows:

- We introduce the first global-scale ML-ready dataset *CarbonGlobe* for forest carbon dynamics monitoring and forecasting at 0.5° resolution, and the temporal range spans over 40 years.

- *CarbonGlobe* covers a comprehensive set of $100^+$ variables integrated from heterogeneous sources, and includes different scenarios for training and testing to resemble diverse application conditions.

- The dataset helps evaluate ML models' ability and build benchmarks for long-term forecasting under different initial conditions and cross-domain generalization (e.g., climate zones).

- We carry out benchmark experiments with a suite of ML forecasting models, using both standard and new metrics designed for the carbon forecasting problem (e.g., delta and cumulative errors).

## 2  Related Work

**Deep learning emulators for theory-based models.**    There have been increasing efforts to develop deep learning emulators for approximating theory-based models in various domains [31, 62]. For weather forecasting, deep emulators have been created to significantly accelerate traditional numerical models that are computationally expensive, allowing global-scale short-term forecasting at much higher resolutions [33, 6]. Similarly, multi-scale climate simulations which deploy smaller-scale and high-resolution simulators nested within host grid columns, have also been emulated using deep learning models for faster approximation [66, 48, 42, 58]. Moreover, physics simulations, which involve solving complex partial differential equations, have been extensively studied using deep learning approximators [44, 51, 29]. Although there have been many well-established deep learning emulators, they are specifically designed only for target problems (e.g., global convolution) and are not suitable for approximating carbon dynamics in forest ecosystems [61]. One recent work [61] proposes the first deep learning emulator for forecasting forest carbon dynamics, but it only focuses on the Northeastern US and does not publish any datasets.

**Time-series forecasting.**    Deep learning models have shown promising performance in time-series forecasting. Variations of recurrent networks and their integration with convolutional networks have been developed to model spatio-temporal dependency [61, 9, 65, 32]. More recently, transformer-based models have been widely used for long-term time-series forecasting due to their ability to model long-range dependency [56, 10, 11]. Many variants were also developed to enhance the forecasting ability, such as the more efficient ProbSparse self-attention in Informer [69], auto-correlation decomposition in Autoformer [63], cross-feature and cross-time dependency modeling in Crossformer [68], etc. DLinear also revisited the potentials of multi-layer perceptrons and showed comparable performance to Transformer-based models on many occasions [67]. A limitation of these models is that they tend to rely on past sequences as inputs in order to make future forecasts, and the performance reduces if only initial conditions are available. In addition, knowledge-guided learning models, which integrate domain knowledge to enhance generalizability, have shown improvements for various problems, such as forest carbon forecasting [61], lake temperature monitoring [27, 26], solar forecasting [34, 36, 35], etc. These models are specifically designed for target domain problems.

**ML-ready datasets for Earth Science.**    There has been increasing attention on creating and sharing high-quality, large-scale, ML-ready datasets for solving Earth science challenges. For example, weather forecasting now benefits from various choices of ML-ready datasets, such as weather conditions in WeatherBench 2 [47], satellite data in EarthNet2021 [49], as well as theory-based climate simulations in ClimSim [66], ClimateSet [28] and ClimART [8]. Similarly, there have been a large number of ML-ready datasets for various tasks in land surface monitoring, including image classification and retrieval in BigEarthNet-MM [54], wildfire detection in Mesogeos [30], crop type identification in CropHarvest [55], building and road detection in SpaceNet [52], etc. Despite the availability of various datasets, no publicly available ML-ready datasets currently exist for global-scale carbon forecasting in forest ecosystems, hindering the development and adoption of ML models for this critical task of carbon neutralization and climate mitigation.

# 3   CarbonGlobe: Dataset Construction

## 3.1   Background: The Ecosystem Demography (ED) model for carbon forecasting

ED is a global demographic, process-based ecosystem model that mechanistically tracks plant dynamics, including growth, mortality, and reproduction, and integrates it with other larger-scale submodules on the carbon cycle, hydrology, and soil biogeochemistry [25, 43, 38]. The carbon cycle tracked includes uptake by photosynthesis, carbon allocation to the growth of biomass (e.g., in leaves, roots, and stems); redistribution of carbon from plants to soil due to dead plants by mortality or disturbance; carbon decomposition, and carbon combustion from fire. ED explicitly considers a variety of drivers including meteorological properties, soil physical properties, and $CO_2$ concentration. The sophisticated processes at individual plant scales are scaled up through a set of partial differential equations governing demographic dynamics. ED outputs a range of ecological variables including carbon stocks (e.g., vegetation carbon, soil carbon), carbon fluxes (e.g., gross primary productivity), water fluxes (e.g., evapotranspiration), and vegetation structure (e.g., canopy height).

**Validation and applications.** ED has been extensively calibrated and validated with real observations using multi-source data from vegetation distribution, vegetation vertical structure, carbon fluxes, airborne and spaceborne LiDAR observations, ground measurements of forest inventory, as well as atmospheric inversions of land net carbon fluxes at both global and regional scales [22, 39, 38, 16]. These evaluations have demonstrated strong alignments between ED outputs and observations from field measurements and satellite observations. More validation information is available in the Appendix. Fig. 1 shows an example where ED outputs showed similar seasonal trends with monthly GPP measurements from in-situ ground monitoring stations in the ABCflux database [57]. Given the high quality of the ED model, its results have been used to support important reports and programs, including the Global Carbon Budget [16, 17], NASA Carbon Monitoring System, and the Department of the Environment in Maryland, US [2, 1, 23].

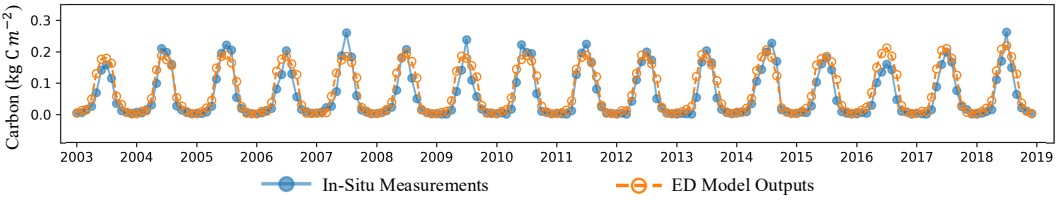

Figure 1: GPP comparisons between the calibrated ED model and in-situ measurements.

## 3.2   Stage 1: Integrating inputs from heterogeneous sources

The first key step is to process and integrate all the inputs needed for the ED model in order to generate the carbon forecasting outputs. As ED considers a diverse range of plant-level and larger-scale processes (e.g., carbon cycle), we follow the established workflow in [38] to prepare three types of information from heterogeneous data sources as detailed in the following: (1) **Meteorological forcing**: The data is collected from NASA's Modern-Era Retrospective analysis for Research and Applications, version 2 (MERRA-2) [18], including surface air temperature, specific humidity, wind speed, precipitation, incident shortwave radiation, and multilayer soil temperature. (2) **$CO_2$ concentration:** The surface $CO_2$ information is collected from the NOAA CarbonTracker Database [46]. $CO_2$ concentration has major impacts on plant physiology such as photosynthesis, water use efficiency, etc. (3) **Soil properties**: Soil data is collected from ROSETTA [41], and the variables include depth, hydraulic conductivity, as well as residual and saturated volumetric water content. Detailed descriptions of the datasets and processing steps are provided in the supplementary document.

**Spatial and temporal coverage.** The inputs are collected at the global scale with a 0.5° spatial resolution, excluding Antarctica. The choice of 0.5° spatial resolution follows established standards in Earth system modeling, where datasets at 0.5°-2° are widely used to advance understanding of the global terrestrial carbon cycle [4, 21, 13]. The resolution also aligns with major international frameworks such as the Global Carbon Budget [16, 17], where ED has been included since 2023. From the AI model development perspective, 0.5° or coarser resolution makes it easier for training

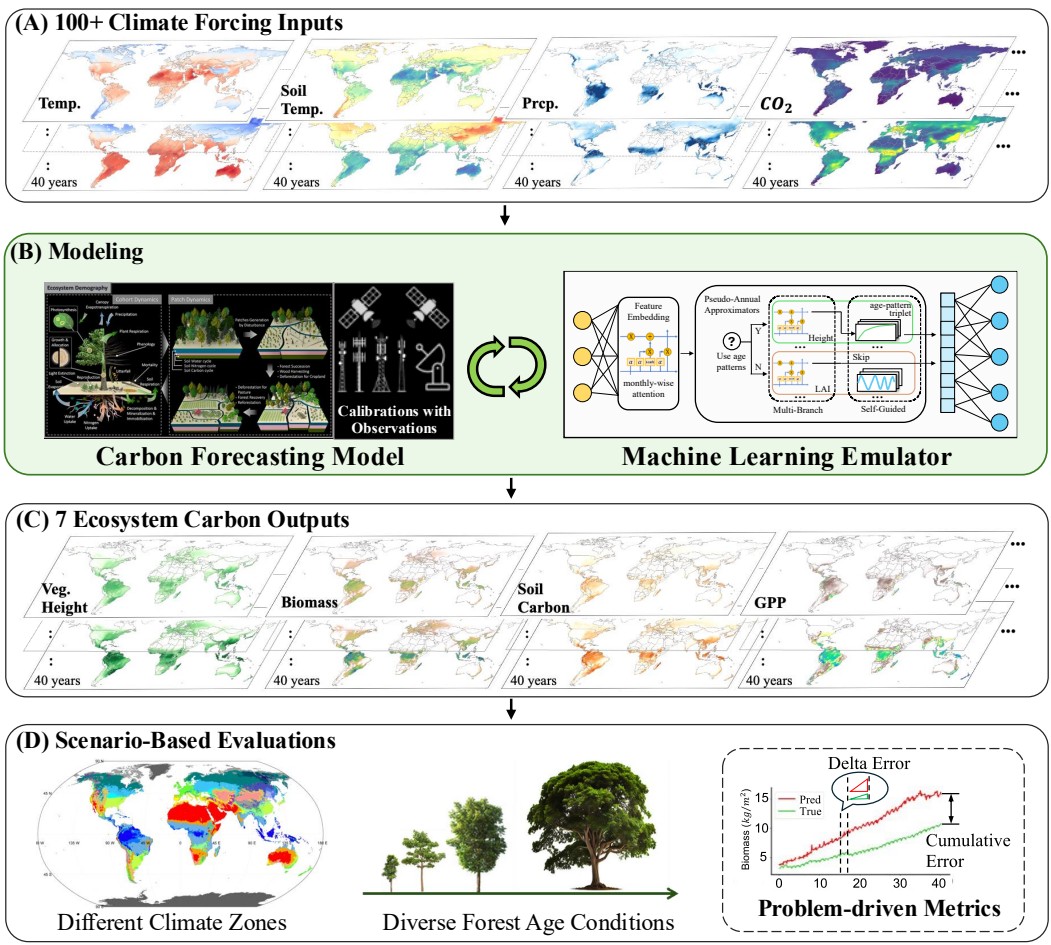

Figure 2: Overview of *CarbonGlobe*. (A) *CarbonGlobe* collects global input data at 0.5° spatial resolution over a 40-year period from heterogeneous sources, including climate forcings, $CO_2$ concentrations, and soil properties. (B) A pioneering theory-based ecosystem model (ED), calibrated with ground network measurements and satellite LiDAR data, takes these inputs to forecast carbon dynamics. Once trained on *CarbonGlobe* inputs and outputs, machine learning emulators replicate this process with significantly reduced computational cost, enabling applications at higher spatial or temporal resolutions. (C) Model outputs include key ecosystem carbon variables at the same resolutions. (D) *CarbonGlobe* defines standardized pipelines for better reflecting downstream domain applications through three evaluation scenarios and two problem-driven evaluation metrics.

and evaluation without an overwhelmingly large data size for general users, which is a common consideration for related datasets such as WeatherBench [47]. For the temporal range, we select a 40-year period (1981-2021) to construct the dataset. The reason that we select this range instead of a future range is to best ensure the quality of the ED outputs in Stage 2, where model calibration can be effectively conducted using available observations. This does not alter the forms of the inputs or outputs for the forecasting task from the ML perspective, as the ML models will start from the initial year in the range and forecast till the end of the sequence, regardless of the time period selected. The next two sections will discuss the output generation using ED and the details of the ML-ready input-output formats for the data.

### 3.3 Stage 2: Generating ED outputs

Given the prepared input data from Stage 1, ED is then executed to forecast carbon dynamics. For the outputs, our carbon experts select the following 7 variables as forecasting targets: vegetation height, aboveground biomass (AGB), soil carbon (SC), leaf area index (LAI), gross primary production

(GPP), net primary production (NPP), and heterotrophic respiration (Rh). These output data have been extensively evaluated against reference datasets and compared between ecosystem models [22, 38, 40].

To generate this dataset, we ran the ED model in our HPC cluster equipped with AMD EPYC Processors for a total of ~812 CPU days. This high computational cost presents a major bottleneck for scaling up to higher spatial resolutions (e.g. will take 25x longer to produce data at 0.1°) or extending the dataset over longer temporal horizons. To address these limitations, we structure the dataset in a ML-ready format in the following section to support the development of deep learning emulators, with the goal of accelerating large-scale, long-term ecosystem carbon forecasting.

### 3.4 Stage 3: Developing ML-ready data formats, scenarios, and problem-specific metrics

The objective of this stage is to complete the generation of the ML-ready dataset *CarbonGlobe*, enabling researchers to efficiently train deep learning emulators for the computationally expensive ED model. To better represent diverse application needs, we also construct three evaluation scenarios for benchmarking and model comparison. Finally, we develop two problem-driven evaluation metrics to better reflect the needs in the downstream applications. The overall dataset is illustrated in Fig. 2.

**ML-ready data formats.** Given the inputs and outputs described in the previous two stages, *CarbonGlobe* presents a global dataset at 0.5° spatial resolution (54,152 land site locations in total) for a time span of 40 years. In addition, ED uses tree age as a seed condition during its theory-based propagation. To make our dataset more comprehensive, we generated the sequences of outputs at each location using 15 different seed tree ages in ED. For clarity, here we formally define the formats of inputs and outputs for the ML forecasting task. The input $\mathbf{X} \in \mathbb{R}^{N \times T \times M \times D}$ covers the time-series of variables that are necessary for predicting the output targets, where $N = n \cdot a$ is the number of site locations $n$ multiplies the number of seed tree ages $a$ (i.e., one time-series sequence per location and tree age); $T$ is the number of years in each sequence; $M$ is the number of time steps per year used to chronologically order inputs within a year (here we have one month as one step); and $D$ is the number of variables at each time step. $M$ can also be merged with $D$ (i.e., $D \cdot M$) if the chronological order is ignored within a year. In total we have $N = 54,152 \times 15 = 812,280$ time-series sequences for the forecasting task, and each sequence covers 40 consecutive years. For each year, there are $M = 12$ monthly sub-steps to provide a chronological order as needed, and at each step, there are $D = 136$ variables from Stage 1 including both dynamic and static variables, where static variables are duplicated for each time step. Note that in ED the sequences are independent of each other, so we do not organize nearby site locations using an image format to keep consistency with the original model. The output $\mathbf{Y} \in \mathbb{R}^{N \times T \times D'}$ has the same $N$ sequences of length $T$ years, and $D'$ output variables for each year. As introduced in Stage 2, there are $D' = 7$ output target variables. Finally, there is also a vector of initial target variables $\mathbf{v} \in \mathbb{R}^{N \times D'}$ that is needed at the very beginning step of the forecasting, and different initial $\mathbf{v}$ lead to different sequences. Finally, the ML task is defined as: Given time-series input $\mathbf{X}$ and the initial $\mathbf{v}$, forecast the output $\mathbf{Y}$ sequentially for all the $T$ steps. As the sequences can be extensively long, it is a common practice during training to split the sequences into smaller segments to reduce memory consumption [50, 31, 61]. In this case, $\mathbf{v}$ for each segment can be extracted from $\mathbf{Y}$ using target variable values from the last time step before the segment.

**Scenarios for evaluation.** As *CarbonGlobe* has large geographic and temporal coverage, there exist heterogeneous patterns of carbon dynamics affected by climate types and forest maturity represented by age. To better understand the performance of ML models under different scenarios, the dataset includes three evaluation scenarios: (1) **Overall evaluation:** This is the standard ML evaluation where all test data are used together to assess model performance. To better align with application needs, we use only 1/16 of the data for training and the rest for testing. The reason is that an important goal of the ML emulators is to reduce the forecasting time and enable capabilities such as forecasting at higher spatial resolution. In practice training samples can be generated using a coarse grid, and then a well-trained emulator can be used to predict all results at a finer grid. In this case, it is equivalent to uniformly sampling a subset of site locations over space for training, where the $1/16$ sampling ratio we used is approximately the same as using a 4x-coarser grid for training. (2) **Climate-based evaluation:** Carbon dynamics are significantly affected by climate properties such as precipitation and temperature. We collect the Köppen-Geiger climate classification map from [5] and process it into the same spatial reference system of the carbon data. Next, we separate test data points based on

the major climate zones, including tropical, arid, temperate, cold, and polar zones. The performance statistics can then be computed separately for each climate zone. (3) **Forest-age-based evaluation:** Forest age influences carbon dynamics in various ways. For example, forests have more rapid carbon sequestration with faster growth at younger ages, which gradually slow down as they mature. Thus, we include this forest-age-based scenario where evaluations are separated by tree ages to understand the corresponding performance variability of ML models.

**Problem-driven evaluation metrics.** Traditional ML time-series metrics (e.g., RMSE, MAE) tend to average errors over all time steps, which may not best reflect error accumulations in long-term forecasting tasks. To address this, we introduce two additional problem-driven evaluation metrics that better reflect ecological and temporal dynamics: **cumulative error** and **delta error**. These metrics were developed in consultation with domain experts in carbon and ecosystem modeling and align more closely with practical monitoring needs. Specifically, **cumulative error** measures the overall error at the very last step (i.e., $T^{th}$ step) of each sequence of length $T$: $E_C = ||\mathbf{Y}_T - \hat{\mathbf{Y}}_T||_2^2$, where $T$ denotes the very last step of each sequence and $\hat{\mathbf{Y}}$ is the prediction by the ML model. **Delta error** represents the error on the changes between two consecutive time steps, and the year-over-year changes are important for monitoring differences caused by variations in environment conditions : $E_\Delta = \sum_{t=2}^{T} ||(\mathbf{Y}_t - \mathbf{Y}_{t-1}) - (\hat{\mathbf{Y}}_t - \hat{\mathbf{Y}}_{t-1})||_2^2/(T-1)$.

## 4    Experiments

### 4.1    Candidate methods

We evaluate the performance of a set of ML methods on *CarbonGlobe* that are broadly used for time-series forecasting, including LSTM variants, transformer-based models, as well as a recent deep emulator designed for ED. In addition, for long-term emulation, training only on original (true) data often leads to faster error accumulation [50]. For example, during training, initial target variables **v** (Sec. 3.4) in shorter segments are based on the true **Y**. However, during testing, the model has to forecast the entire sequence relying on the initial $\mathbf{Y}_0$, which means **v** in the middle steps will be purely based on the predicted $\hat{\mathbf{Y}}$, resulting in higher error accumulation. To mitigate this, we integrate the strategy by [50, 31, 53] to insert perturbations to initial target variables **v** by adding random-walk noise $\mathcal{N} = (0, \sigma)$ with a small standard deviation (e.g., $\sigma = 1e^{-4}$). This strategy shared by emulators is largely problem-agnostic and significantly reduces error accumulation for all candidate methods. Thus, we apply it by default to all the following methods (results without this strategy and details of model training parameters and architecture are included in the supplementary document):

- **LSTM:** A standard LSTM taking time-series ED inputs and outputting targets [19].
- **LSTNet:** A time-series model extracting short-term dependencies using convolution along time-feature dimension and long-term temporal patterns using RNN [32].
- **DeepED:** An LSTM-based deep learning emulator for ED with specialized designs on error accumulation reduction and knowledge-guided learning [61].
- **Transformer (TF):** A vanilla transformer model with a self-attention mechanism to capture dependencies across all time steps during the forecasting [56].
- **Informer (IF):** A transformer variant with a ProbSparse attention for reduced complexity [69].
- **DLinear (DL):** A linear decomposition model to separate data into trend and seasonality, showing comparable performance among transformer-based models while maintaining efficiency [67].
- **Crossformer (CF):** A transformer variant with a two-stage attention mechanism for modeling both cross-time and cross-feature dependencies [68].
- **TimeXer (TX):** A transformer variant with a separated modeling strategy to capture both inter-target relationships and input-target dependencies [59].

### 4.2    Results

**Overall evaluation.** Table 1 shows the overall forecasting performance on target variables using 4 evaluation metrics. The ED emulator, DeepED, outperforms other models on target variables height, AGB, SC and LAI, while Transformer (TF) and Informer (IF) have the best or the second-best performance on NPP, GPP, and Rh. The interesting separations of the best models reflect the

Table 1: Overall performance of ML methods in RMSE, MAE, delta (Δ), and cumulative error (CE).

| Model | Height | | AGB | | SC | | LAI | | GPP | | NPP | | Rh | |
|---|---|---|---|---|---|---|---|---|---|---|---|---|---|---|
| | RMSE | MAE | RMSE | MAE | RMSE | MAE | RMSE | MAE | RMSE | MAE | RMSE | MAE | RMSE | MAE |
| LSTM | 3.352 | 1.890 | 1.363 | 0.869 | 1.942 | 1.354 | 0.708 | 0.460 | 0.281 | 0.137 | 0.139 | 0.068 | 0.161 | 0.088 |
| LSTNet | 2.887 | 1.608 | 1.228 | 0.798 | 1.284 | 0.833 | 0.661 | 0.427 | 0.270 | 0.129 | 0.135 | 0.064 | 0.149 | 0.081 |
| DeepED | **1.900** | **0.891** | **0.491** | **0.248** | **0.660** | **0.354** | **0.416** | **0.210** | 0.258 | 0.128 | 0.127 | 0.059 | 0.148 | 0.084 |
| TF | 2.888 | 1.750 | 1.154 | 0.626 | 1.710 | 1.083 | 0.449 | 0.247 | **0.217** | **0.088** | **0.107** | **0.043** | **0.121** | **0.059** |
| IF | 2.915 | 1.773 | 1.266 | 0.654 | 1.803 | 1.196 | 0.461 | 0.250 | 0.234 | 0.094 | 0.116 | 0.045 | 0.124 | 0.063 |
| DL | 5.731 | 4.124 | 3.191 | 2.284 | 4.371 | 3.383 | 1.119 | 0.814 | 0.578 | 0.389 | 0.289 | 0.193 | 0.268 | 0.186 |
| CF | 2.910 | 1.563 | 1.218 | 0.614 | 1.144 | 0.637 | 0.506 | 0.298 | 0.235 | 0.114 | 0.117 | 0.056 | 0.131 | 0.065 |
| TX | 3.879 | 2.301 | 1.583 | 1.046 | 1.770 | 1.317 | 1.127 | 0.701 | 0.972 | 0.477 | 0.581 | 0.237 | 0.512 | 0.284 |
| | Δ | CE | Δ | CE | Δ | CE | Δ | CE | Δ | CE | Δ | CE | Δ | CE |
| LSTM | 0.433 | 4.794 | 0.183 | 1.660 | 0.188 | 2.455 | 0.413 | 0.838 | 0.289 | 0.325 | 0.145 | 0.160 | 0.179 | 0.177 |
| LSTNet | 0.421 | 4.003 | 0.189 | 1.454 | 0.150 | 1.881 | 0.404 | 0.776 | 0.280 | 0.309 | 0.142 | 0.154 | 0.166 | 0.165 |
| DeepED | **0.398** | **2.424** | **0.114** | **0.604** | **0.120** | **0.955** | 0.377 | **0.475** | 0.244 | 0.299 | 0.123 | 0.143 | 0.151 | 0.187 |
| TF | 0.419 | 3.925 | 0.131 | 1.592 | 0.148 | 2.525 | 0.287 | 0.552 | 0.169 | 0.253 | **0.085** | **0.125** | 0.113 | **0.148** |
| IF | 0.420 | 3.966 | 0.135 | 1.749 | 0.150 | 2.693 | **0.284** | 0.570 | **0.168** | 0.271 | **0.085** | 0.134 | **0.111** | 0.153 |
| DL | 0.867 | 5.905 | 0.284 | 3.457 | 0.348 | 4.404 | 0.428 | 1.270 | 0.334 | 0.675 | 0.168 | 0.336 | 0.188 | 0.309 |
| CF | 0.413 | 3.904 | 0.144 | 1.616 | 0.139 | 1.770 | 0.299 | 0.640 | 0.183 | 0.285 | 0.093 | 0.142 | 0.129 | 0.159 |
| TX | 0.423 | 5.634 | 0.155 | 2.328 | 0.146 | 2.713 | 0.423 | 1.898 | 0.344 | 2.009 | 0.180 | 1.430 | 0.192 | 0.920 |

*__Bold__ = best model, Underline = runner-up.

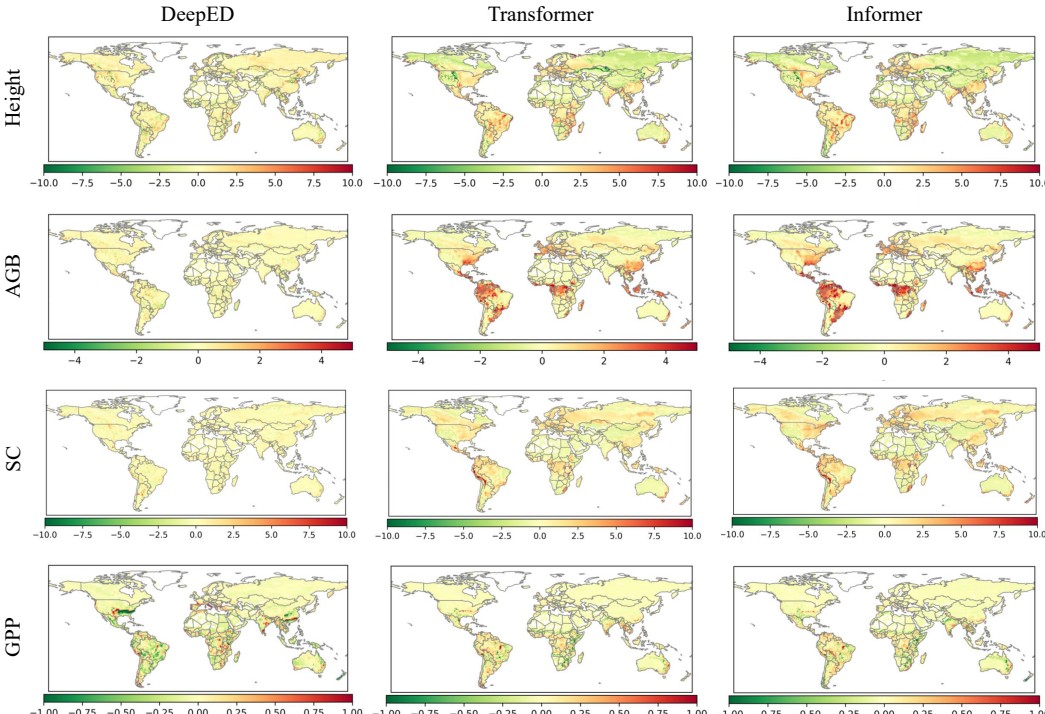

Figure 3: Global distribution of the difference between the results of emulators and ED in Year 40.

heterogeneous patterns of different target variables. In particular, height, AGB, SC and LAI tend to have an increasing trend over time, where current status has stronger effects on the forecasting of the next time step. In contrast, NPP, GPP, and Rh have stronger seasonality, where variations of climate and soil conditions have a potentially higher influence on forecasting. Transformer-based architectures are able to capture complex patterns through the attention mechanism, which potentially contributes to their better performance on these variables. Crossformer also receives many second-best performances but does not show advantages over Transformer and Informer on this problem. Other models, such as DLinear, TimeXer, and LSTM, have large errors for long-term forecasting without targeted designs based on the ED model. Two proposed evaluation metrics, delta error and cumulative error (Sec. 3.4), also reflect the new insights. For example, the relative differences between the methods are reduced – sometimes with the ranking changed – when moving from RMSE and MAE to the delta errors. In addition, variables such as height, AGB, SC, and LAI exhibit larger

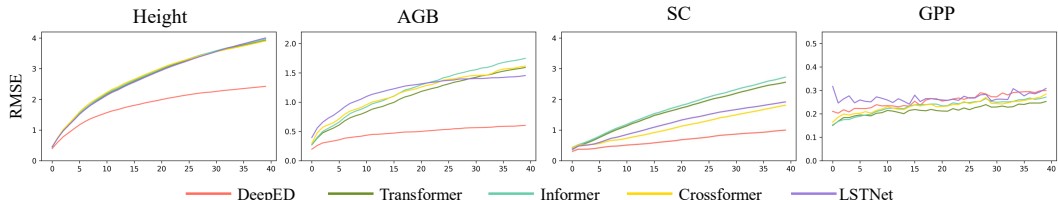

Figure 4: Visualization along the temporal dimension: RMSE over 40 years.

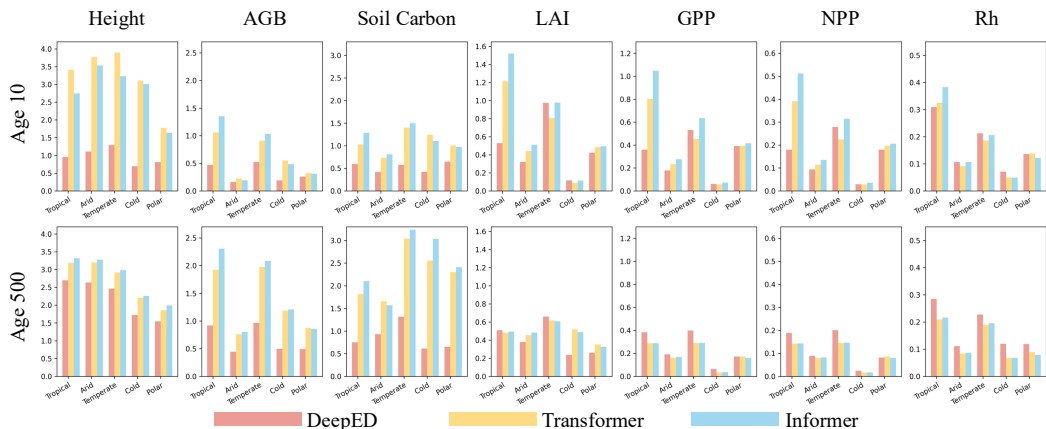

Figure 5: Error distributions among different climate zones and forest ages.

changes in error patterns, while GPP, NPP, and Rh show relatively minor differences. This also reflects the impact of variable characteristics (e.g., monotonic, periodical) on error patterns.

**Patterns over space and time.** Fig. 3 and Fig. 4 highlight the spatial and temporal variations of prediction quality. Fig. 3 shows the error distribution (prediction minus label) over the globe at the final forecasting step. We use the three best-performing models and three representative variables as examples, and more results are included in the supplementary document. Through the maps, we can see clear patterns of spatial clusters. DeepED tends to slightly overestimate height, AGB, and SC in regions with sparse forest coverage or colder climates, while Transformer and Informer show overestimation of these variables in densely forested areas such as the Eastern US, Central Africa, as well as the Amazon region. Both Transformer and Informer exhibit an interesting pattern: they underestimate height but overestimate AGB in the western Siberia region. While these variables are typically correlated, this discrepancy suggests a region-specific divergence in their relationship within Siberia. For temporal patterns in Fig. 4, DeepED shows the best performance in predicting height, AGB and SC with the smallest errors over 40 years. In comparison, Transformer has the best overall performance in predicting GPP. While the separation between the methods is less significant for GPP, LSTM-based methods overall have higher errors for this variable.

**Scenarios in climate zones and forest ages.** Fig. 5 shows the results under different climate zones and forest ages to better understand the model performances in different scenarios. Interestingly, we do observe significant variations caused by both climate and forest ages, as well as their interplay. First, climate-induced differences have stronger expressions in certain variables such as AGB and GPP. For example, AGB errors are higher in tropical regions than in colder climates, likely due to differences in forest structure. In terms of forest age, DeepED in general outperforms Transformer at smaller seed forest ages. This pattern also holds for height, AGB and soil carbon at larger seed ages, while Transformer starts to outperform on GPP, NPP and Rh. These findings provide insights for future model development that incorporates more heterogeneity-awareness or knowledge-guided methods [64] to better capture region- and condition-specific dynamics.

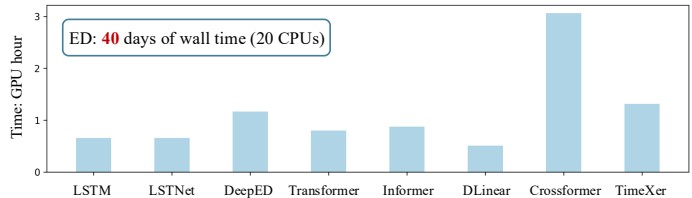

Figure 6: Inference time of ML models.

**Execution time.** Fig. 6 shows the inference time of the ML-based emulators for forecasting global carbon dynamics over 40 years using a single RTX A4500 GPU. For context, the same computation took ED about 40.6 days of wall time using 20 CPU cores in parallel, equivalent to roughly 812 CPU days in total. In comparison, the ML models are able to reduce the forecasting time by orders of magnitude (e.g., about 1-1.5 hours for most models). DeepED has a slightly longer inference time due to multi-scale architecture designed to reduce error accumulation and handle heterogeneous variables. DLinear has the shortest computation time, but its prediction quality is less accurate compared to the other candidate methods in the evaluation.

## 5 Conclusion, Limitations and Future Work

This paper presents a new global-scale benchmark dataset *CarbonGlobe* for carbon forecasting at $0.5°$ spatial resolution spanning 40 years. The dataset integrates $100^+$ variables from heterogeneous sources and is constructed in a ML-ready format. For performance benchmarking we also developed a suite of time-series forecasting methods and evaluated them using both standard and problem-driven metrics under various scenarios such as different climate zones and forest conditions. *CarbonGlobe* will facilitate the development of ML models for scalable and long-term carbon forecasting.

**Limitations, opportunities, and future work.** While *CarbonGlobe* is the first global-scale ML-ready dataset for forest carbon dynamics, we would like to acknowledge several limitations of it. First, while the ED model itself is open-sourced, running ED still requires significant domain expertise, making it difficult for non-experts to generate additional data under new conditions (e.g., higher-resolution, local-scale forecasting). Moreover, the current dataset has not incorporated simulations for future (e.g., from 2025 to 2100 using conditions from CMIP-6). We plan to add future simulation runs to the dataset as additional test cases for the emulators. Second, the current task setup does not cover emerging ML paradigms such as active learning and knowledge-guided learning, or models not based on deep learning. We plan to extend the dataset with new task formulations to support these directions. Third, intermediate variables and submodule outputs from ED (e.g., hydrology, fire, or sub-annual carbon fluxes) are not included due to storage constraints, but future versions may incorporate them to support more physically-consistent model development. Finally, we plan to expand the integration of alternative forecasting strategies, such as non-autoregressive formulations, which can potentially help further reduce error accumulation and improve model performance.

## Acknowledgments

Zhihao Wang, Yiqun Xie, Ruohan Li, Zhili Li, and Shuo Xu are supported in part by the NSF under Grant No. 2126474, 2147195, 2425844, and 2530610; NASA under grant 80NSSC25K0013 and 80NSSC25K7221; Google's AI for Social Good Impact Scholars program; and the Zaratan cluster at the University of Maryland. George Hurtt is supported by the NASA CMS under Grant No. 80NSSC25K7221 and NASA EIS under Grant No. 80NSSC22K1733. Lei Ma is supported by the NASA ECIPES under Grant No. 80NSSC24K1632. Xiaowei Jia was partially supported the NSF under Grant No. 2239175, 2147195, 2316305, 2425845, 2530609, 2203581; NASA under Grant No. 80NSSC24K1061 and 80NSSC25K0013; the USGS awards G21AC10564 and G22AC00266; and Pitt Momentum Funds and CRC at the University of Pittsburgh. Yanhua Li was partially supported by NSF IIS-1942680 (CAREER) and CNS-1952085. We would like to acknowledge high-performance computing support from the Derecho system (doi:10.5065/qx9a-pg09) provided by the NSF National Center for Atmospheric Research (NCAR), sponsored by the National Science Foundation.

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

# Appendix

## A    Dataset: Access and Additional Details

### A.1    Dataset Access

The datasets are available for download at the following link for review purposes. All datasets have been uploaded to Kaggle and will be made publicly accessible upon acceptance of the paper. The dataset can be assessed from `https://www.kaggle.com/datasets/zhihaow/carbonglobe`.

### A.2    Dataset Variables and Statistics

In the inputs, *CarbonGlobe* covers a comprehensive set of variables integrated from heterogeneous sources, including meteorological properties, soil physical properties, and $CO_2$ concentration. In the outputs, *CarbonGlobe* provides a range of ecological variables including carbon stocks (e.g., vegetation carbon, soil carbon), carbon fluxes (e.g., gross primary productivity), and vegetation structure (e.g., canopy height). Table 2 lists all input and output variables followed by summarized data descriptions.

In Fig. 7, we present statistical information on output variables along forest growth to provide a comprehensive understanding of the central tendency and variability of the output variables across different stages of forest development. We also provide detailed distributions for all input variables in our dataset. All data statistics are calculated across the globe over 40 years and visualized based on the age of the forest. This provides insights into the trend of each variable. For example, height has a steady increase during early ages and then becomes more stable whereas the above-ground biomass (AGB) continues to increase as the tree crown grows larger in volume and higher in density.

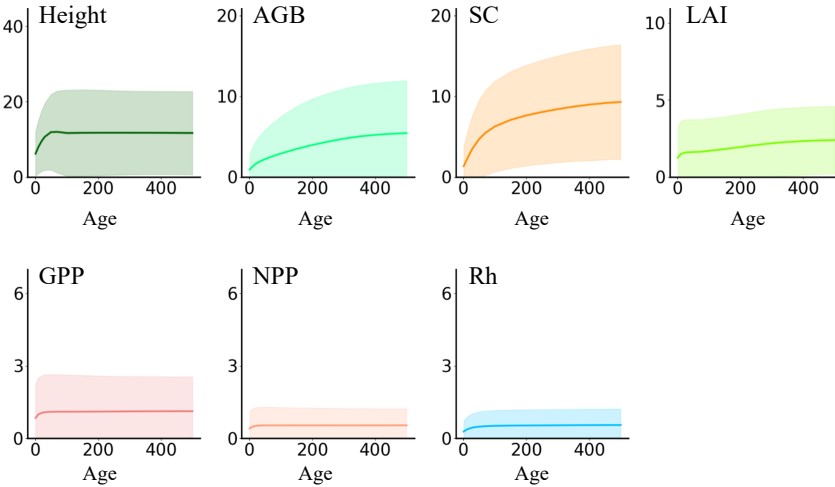

Figure 7: Statistical information on the mean (solid line) and standard deviation (shaded areas) of output variables in different forest ages.

### A.3    Data Collection and Preprocessing

This section presents detailed information about data collection, access licences, and preprocessing steps.

- **Meteorological data**: The data from NASA's Modern-Era Retrospective analysis for Research and Applications, version 2 (MERRA-2) [18] was collected from `https://gmao.gsfc.nasa.gov/reanalysis/MERRA-2/` using official data download website MDISC, managed by the NASA Goddard Earth Sciences Data and Information Services Center. There are no restrictions on the use of data except for referencing the original paper `https://gmao.gsfc.nasa.gov/reanalysis/MERRA-2/citing_MERRA-2/`. After the download, all variables were aggregated to the monthly average for each year.

Table 2: Overview of input and output variables (first and second columns) of *CarbonGlobe* dataset. The subsequent columns detail the variable name, feature dimension, units, and brief descriptions. More details about the theory-based model can be found in [38].

| In | Out | Variable | Count$^\dagger$ | Unit | Description$^\ddagger$ |
|----|-----|----------|-------|------|-------------|
| ✓ | | ta_m | 1 | K | Monthly averaged air temperature |
| ✓ | | pr | 1 | mm | Total monthly precipitation |
| ✓ | | tsl1 | 1 | K | Monthly averaged soil temp. at 0.0988m |
| ✓ | | tsl2 | 1 | K | Monthly averaged soil temp. at 0.1952m |
| ✓ | | tsl3 | 1 | K | Monthly averaged soil temp. at 0.3859m |
| ✓ | | tsl4 | 1 | K | Monthly averaged soil temp. at 0.7626m |
| ✓ | | tsl5 | 1 | K | Monthly averaged soil temp. at 1.5071m |
| ✓ | | tsl6 | 1 | K | Monthly averaged soil temp. at 10m |
| ✓ | | co2 | 24 | ppm | Monthly average of hourly $CO_2$ ambient |
| ✓ | | dst | 1 | 1 | Disturbance rate |
| ✓ | | hus | 24 | 1 | Monthly average of hourly air specific humidity |
| ✓ | | ta_h | 24 | K | Monthly average of hourly air temperature |
| ✓ | | rsds | 24 | W m$^{-2}$ | Monthly average of hourly DSR |
| ✓ | | sfcWind | 24 | m s$^{-1}$ | Monthly average of hourly wind speed |
| ✓ | | k_sat | 1 | mm yr$^{-1}$ | Saturated hydraulic conductivity |
| ✓ | | s_theta | 1 | m$^3$ m$^{-3}$ | Saturated water content in MVG |
| ✓ | | r_theta | 1 | m$^3$ m$^{-3}$ | Residual water content in MVG |
| ✓ | | L | 1 | 1 | Parameter L in MVG |
| ✓ | | n | 1 | 1 | Parameter n in MVG |
| ✓ | | m | 1 | 1 | Parameter m in MVG |
| ✓ | | sd | 1 | mm | Soil depth to bedrock |
| ✓* | ✓ | height | 1 | m | Vegetation canopy height |
| ✓* | ✓ | agb | 1 | kgC m$^{-2}$ | Aboveground biomass |
| ✓* | ✓ | sc | 1 | kgC m$^{-2}$ | Soil carbon |
| ✓* | ✓ | lai | 1 | 1 | Leaf area index |
| ✓* | ✓ | gpp | 1 | kgC m$^{-2}$ | Annual gross primary production |
| ✓* | ✓ | npp | 1 | kgC m$^{-2}$ | Annual net primary product |
| ✓* | ✓ | rh | 1 | kgC m$^{-2}$ | Annual heterotrophic respiration |

*It is part of the input only at the beginning of the step, which is the same for the ED model. During forecasting, the output from the previous step is used as the input for the next step.

$^\dagger$The number of sub-variables. 24 means the values are averaged for each hour of the day.

$^\ddagger$Abbreviations: Mualem–van Genuchten equations (MVG), which are used to quantify the hydraulic properties of unsaturated soils; downward solar radiation (DSR); temperature (temp.).

- **$CO_2$ Data:** The surface $CO_2$ information from the NOAA CarbonTracker Database [46] was collected from `https://gml.noaa.gov/ccgg/carbontracker/` using the data archive system provided on the website directly. There are no restrictions on the use of data except for referencing the publications `https://gml.noaa.gov/ccgg/carbontracker/CT2007/citation.php`. The $CO_2$ data was first linearly interpolated from $3° \times 2°$ to $0.5° \times 0.5°$ at the spatial scale and from 3h to hourly at the temporal scale.

- **Soil properties**: Soil data from ROSETTA [41] was collected from `https://doi.pangaea.de/10.1594/PANGAEA.870605` under the Creative Commons Attribution 3.0 License. The soil data was originally delivered for latitudes from 60°S to 90°N, excluding Antarctica. We aggregated the data from $0.25° \times 0.25°$ to $0.5° \times 0.5°$.

- **Climate Classification Data**: We collected the Köppen-Geiger climate classification map [5] from `https://figshare.com/articles/dataset/Present_and_future_K_ppen-Geiger_climate_classification_maps_at_1-km_resolution/6396959/2` under the Creative Commons Attribution 4.0 International License. We aggregated 30 climate classification subtypes into 5 major types, based on which we separated our test data points to evaluate the model performance for each climate type. Fig. 8 visualizes the 5 primary climate zones and the corresponding percentage of test data points under each climate condition.

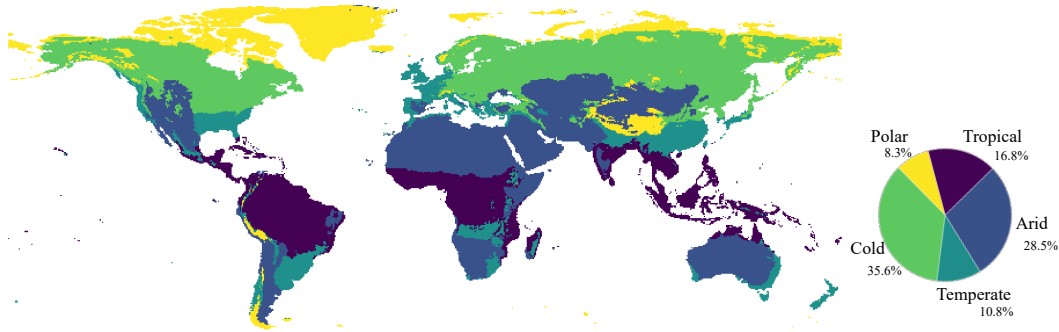

Figure 8: A visualization of the present Köppen-Geiger climate classification map with five major climate zones and the corresponding test sample sizes.

## A.4 Author Statement

The authors of this paper bear full responsibility for any potential violations of rights arising from the data collection included in this research.

## B Models: Access and Additional Details

### B.1 Code Access

We have uploaded the benchmark model code to GitHub, along with detailed documentation on how to run the code and references to the original model papers.. The code can be accessed at: `https://github.com/zhwang0/carbon-globe`.

### B.2 Model Implementation Details

**LSTM.** A standard Long Short-Term Memory (LSTM) model, which is a type of recurrent neural network designed to better handle long-term dependencies [19]. LSTM captures and retains temporal patterns using gating mechanisms, including input, output, and forget gates. In experiments, we used a single LSTM layer with 256 neurons, followed by a linear output layer for targets.

**LSTNet.** Long- and Short-term Time-series network (LSTNet) combines both convolutional and recurrent neural networks for multivariate time series forecasting [32]. Specifically, LSTNet uses CNN to extract short-term patterns in the combined time and feature dimensions (i.e., $T \times D$-shape inputs for the CNN layer where $T$ is the number of time steps and $D$ is the number of features) and then uses RNN to capture longer-term temporal patterns for time-series trends. We implemented LSTNet using a convolutional layer with 256 neurons and a kernel size of (6, 136), a LSTM layer with 256 neurons, and a linear output layer.

**DeepED.** DeepED is a LSTM-based deep learning emulator for the theory-based Ecosystem Demography model for carbon forecasting, with specialized designs on error accumulation reduction and knowledge-guided learning [61]. DeepED has a multi-scale multi-branch structure and de-sequencing loss to reduce error accumulation in long-term forecasting, and proposes a self-guided strategy and significance-based network partitioning to handle heterogeneous temporal patterns among target variables. During the implementation, we used 256 neurons for all LSTM layers in both shared and branched architectures (i.e. all targets have one shared LSTM layer and 3 individual LSTM layers). In the output block in each branch, we used 256, 64, and 1 neurons for three stacked linear layers, respectively.

**Informer.** Informer is designed to address the challenges of long-term time-series forecasting by improving the efficiency and effectiveness of the Transformer architecture [69]. Informer proposes a ProbSparse self-attention mechanism, which only selects the top informative components of a sequence, significantly reducing the computational complexity from $O(L^2)$ to $O(L \log L)$. We used

the default setting of Informer to conduct experiments. Specifically, we used 2 encoder blocks and 1 decoder block, 512 as the dimension of feature embedding outputs, 2048 as the dimension of feed-forward layers, as well as 8 as the number of heads in multi-head attentions. We used GELU as the activation function and 0.05 as the dropout rate.

**Transformer.**   Transformer is a popular neural network with a self-attention mechanism [56], which allows the model to capture the relationships between all elements of a sequence and makes it capture long-term dependencies more effectively than traditional recurrent neural networks. We implemented Transformer by following the same parameter settings in Informer, except that the self-attention mechanism was replaced with the regular full attention, instead of the ProbSparse attention.

**DLinear.**   As more transformer-based models have been widely designed in time-series forecasting, a linear decomposition model (DLinear) shows comparable performance among transformer-based models while maintaining computational efficiency [67]. Specifically, DLinear uses a straightforward linear decomposition strategy to separate the raw time-series data into trend and seasonal components, and model each component using linear transformation to reduce complexity and enhance interpretability. We set 12 months as the temporal steps for modelling trends and seasonality and 1 as the end-of-year step for outputs, followed by a linear projection layer.

**Crossformer.**   Crossformer is a transformer variant with a two-stage attention mechanism for capturing both cross-time and cross-feature dependencies [68]. While traditional transformer-based models primarily focus on temporal dependency, Crossformer further adopts the attention mechanism in the feature dimension to better capture internal relationships. In addition, Crossformer also introduces segmented temporal sequences and a hierarchical encoder-decoder structure to improve model performance. Due to memory constraints, we keep all parameters the same as default except for changing the model size (the dimension used for embedding output) from 256 to 128 and the number of encoder blocks from 3 to 2.

**TimeXer.**   TimeXer is a Transformer-based architecture designed to enhance time series forecasting by explicitly integrating exogenous drivers (e.g. external climate forcings) that influence the target variables (e.g. carbon changes) [59]. It employs a dual-attention mechanism: patch-wise self-attention captures temporal dependencies within the endogenous (target) series, while variate-wise cross-attention models interactions between exogenous and endogenous variables. A global endogenous token serves as a bridge, facilitating the flow of information from exogenous inputs to the target series. We adopt the default configuration used in TimeXer's original long-term forecasting task (i.e. ETTH1), but reduce model size by using 1 encoder layer and setting the model dimension to 1024, due to its more complicated architecture and memory constraints.

## B.3   Model Training Details

In the training stage, we uniformly sampled 1/16 of locations as training data, and the rest for testing. Within training, we used 10% of random samples as the validation set. The rationale behind the uniform sampling is that a key object of ML emulators is to reduce forecasting time and enable capabilities such as forecasting at higher spatial resolution. In practice, this involves generating training samples using a coarse grid, and then employing a well-trained emulator to predict results at a finer grid. Given this context, it is equivalent to uniformly sampling a subset of site locations across the spatial domain for training. The $1/16$ sampling ratio we used is approximately the same as using a 4x-coarser grid for training. Moreover, uniform sampling leads to a more representative coverage of the geographic space, which reduces the risk of the model being spatially overfitted to a sub-region. Practically, as users can choose where to generate the training samples with ED, uniform sampling is also a feasible option. Based on the training samples, we randomly select 10% from them as validation data.

All benchmarking models were trained using the Adam optimizer with an initial learning rate of $10^{-4}$ for up to 60 epochs and a batch size of 1,200 samples across all experiments. Early stopping with a patience of 5 epochs was applied based on validation loss to prevent overfitting. The best-performing model was selected as the one with the lowest validation loss. We observed that the relative ranking of model performance is generally consistent across different hyperparameter settings. For example, in the vegetation height forecasting task, the Transformer's RMSE showed a 0.038 standard deviation

Table 3: Overall performance of ML methods without noise perturbations, evaluated using RMSE, MAE, delta ($\Delta$), and cumulative error (CE).

| Model | Height | | AGB | | SC | | LAI | | GPP | | NPP | | Rh | |
|---|---|---|---|---|---|---|---|---|---|---|---|---|---|---|
| | RMSE | MAE | RMSE | MAE | RMSE | MAE | RMSE | MAE | RMSE | MAE | RMSE | MAE | RMSE | MAE |
| LSTM | 3.376 | 1.906 | 1.511 | 1.007 | 1.691 | 1.049 | 0.709 | 0.459 | 0.272 | 0.133 | 0.139 | 0.067 | 0.160 | 0.090 |
| LSTNet | 2.852 | 1.560 | 1.062 | 0.637 | 1.205 | 0.751 | 0.652 | 0.414 | 0.266 | 0.126 | 0.133 | 0.063 | 0.151 | 0.081 |
| DeepED | **1.880** | **0.856** | **0.486** | **0.244** | **0.684** | **0.333** | **0.435** | **0.228** | 0.253 | 0.122 | 0.125 | 0.060 | 0.199 | 0.085 |
| TF | 2.888 | 1.750 | 1.214 | 0.660 | 1.756 | 1.132 | 0.470 | 0.250 | 0.235 | 0.090 | 0.117 | 0.044 | 0.124 | 0.061 |
| IF | 2.982 | 1.818 | 1.298 | 0.690 | 1.867 | 1.193 | 0.466 | 0.256 | 0.234 | 0.093 | 0.116 | 0.045 | 0.125 | 0.061 |
| DL | 5.835 | 4.230 | 3.323 | 2.383 | 4.537 | 3.514 | 1.161 | 0.843 | 0.599 | 0.401 | 0.297 | 0.197 | 0.276 | 0.190 |
| CF | 3.120 | 1.815 | 1.399 | 0.704 | 1.222 | 0.687 | 0.507 | 0.279 | 0.243 | 0.110 | 0.122 | 0.056 | 0.133 | 0.063 |
| TX | 3.949 | 2.367 | 1.579 | 1.056 | 1.805 | 1.350 | 1.194 | 0.686 | 1.122 | 0.462 | 0.549 | 0.225 | 0.488 | 0.229 |
| | $\Delta$ | CE | $\Delta$ | CE | $\Delta$ | CE | $\Delta$ | CE | $\Delta$ | CE | $\Delta$ | CE | $\Delta$ | CE |
| LSTM | 0.435 | 4.805 | 0.177 | 1.792 | 0.180 | 2.569 | 0.407 | 0.863 | 0.276 | 0.311 | 0.146 | 0.159 | 0.176 | 0.175 |
| LSTNet | 0.417 | 3.948 | 0.165 | 1.375 | 0.147 | 1.827 | 0.401 | 0.774 | 0.276 | 0.299 | 0.138 | 0.148 | 0.169 | 0.167 |
| DeepED | **0.399** | **2.411** | **0.113** | **0.595** | **0.119** | **1.011** | 0.374 | **0.513** | 0.238 | 0.290 | 0.119 | 0.140 | 0.148 | 0.329 |
| TF | 0.419 | 3.926 | 0.135 | 1.708 | 0.156 | 2.680 | 0.286 | 0.594 | 0.169 | 0.280 | 0.085 | 0.139 | 0.113 | 0.154 |
| IF | 0.422 | 4.112 | 0.135 | 1.806 | 0.155 | 2.785 | **0.284** | 0.565 | **0.167** | **0.267** | **0.084** | **0.131** | **0.111** | **0.152** |
| DL | 0.983 | 5.851 | 0.281 | 3.438 | 0.381 | 4.393 | 0.433 | 1.262 | 0.329 | 0.682 | 0.165 | 0.339 | 0.187 | 0.310 |
| CF | 0.419 | 4.265 | 0.147 | 1.966 | 0.141 | 1.923 | 0.317 | 0.634 | 0.190 | 0.295 | 0.097 | 0.148 | 0.126 | 0.162 |
| TX | 0.424 | 5.787 | 0.153 | 2.320 | 0.146 | 2.798 | 0.419 | 2.130 | 0.348 | 2.654 | 0.175 | 1.306 | 0.189 | 0.990 |

***Bold** fonts for best models and underlines for runner-ups.

of RMSE across batch sizes of 400, 800, and 1200; and a 0.014 standard deviation of RMSE for learning rates 0.0001 and 0.001. All models were trained using the standard L2 loss function in the training and tested the model performance under 3 evaluation scenarios using 4 evaluation metrics. The evaluation scenarios include overall evaluation, climate-based evaluation, and forest-age-based evaluation. The evaluation metrics are RMSE, MAE, delta error, and cumulative error.

## C   Additional Results

Due to the page limit in the main paper, we provide additional results here to demonstrate the comprehensive performance of ML models from different perspectives.

**Overall performance without noise perturbation.**    Table 3 presents additional results of model performance without noise perturbation. As we can observe, the table exhibits similar patterns as that from the main paper, where DeepED, Transformer and Informer are the top 1 models for all target variables. More importantly, over 76% of results for the cumulative error with noise-augmented training are better than those without the noises in Table 3. This shows the importance of noise-perturbation during training to reduce error accumulation, by better representing the scenario faced in forecasting where only the initial conditions at the very beginning of the sequence are available. Thus, we recommend using the data perturbation strategy.

**Spatio-temporal patterns of remaining target variables.**    Fig. 9 and Fig. 10 illustrate the spatial and temporal error patterns for the remaining four variables. The maps reveal similar patterns in model performance. DeepED tends to underestimate LAI in some forest regions, while Transformer and Informer show overestimations. This underestimation trend for DeepED is also observed in NPP and Rh, with both Transformer and Informer more likely to overestimate these variables in tropical regions. For temporal patterns shown in Fig. 10, while DeepED has the lowest prediction error in LAI in the last 30 years, the overall performance of other ML models remains relatively similar, except for LSTNet. In comparison, Transformer and Informer exhibit comparable top performance in NPP and Rh predictions. Although the distinction between models is less significant for NPP and Rh, LSTNet generally displays higher errors for these two variables.

**Remaining scenarios in climate and forest ages.**    We further report results across 13 forest-age groups to assess model robustness under varying ecological conditions in Fig 12 and 11. The overall variations introduced by both climate and forest areas are consistent with the results in the main paper. For forest ages, a more detailed trend can be observed: as forests grow from young to mature, the errors for height, AGB, soil carbon, and LAI initially increase and then decrease. In contrast, the errors for GPP and NPP decrease over time, while Rh shows a slight increase in errors according to the distributions.

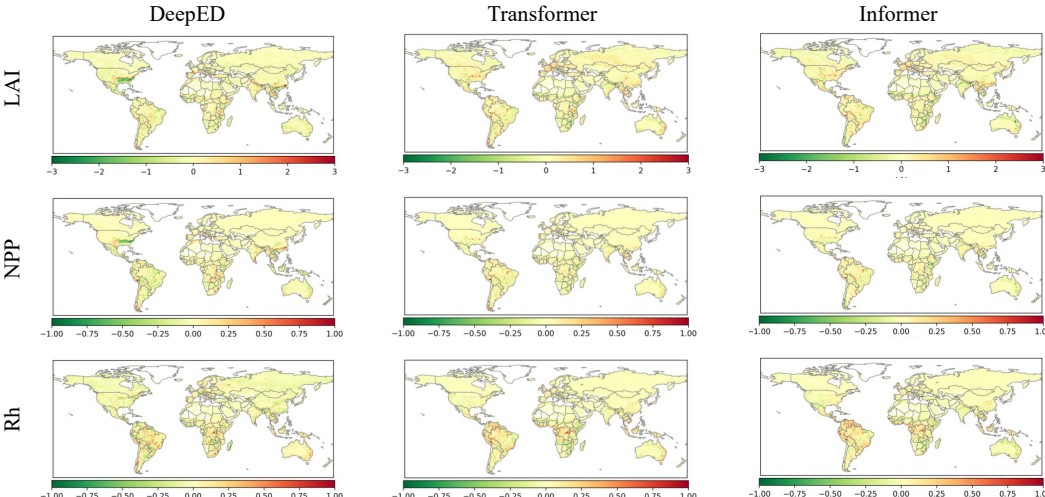

Figure 9: Global distribution of the difference between the results of emulators and ED in Year 40.

**Additional validation information.** We have extensively evaluated ED results in our previous works against observations, independent datasets, and atmospheric inversions. The validation procedures and details can be found in [38, 40, 22]. The process-based model has pioneering ability in integrating observations as initial conditions and inputs. Given its high quality, ED has been adopted and used by major projects such as the NASA Global Carbon Monitoring System, the Global Carbon Budget [16, 17], and the Department of the Environment in Maryland, US [2, 1, 23]. Here we also used the ABoVE GPP in-situ observations to generate some examples. After filtering based on data quality flags, we used the observations from 20 monitoring sites in the North America at year end to align with our dataset's time steps and make comparisons. In this example, both the ED model and Transformer emulator showed an RMSE smaller than 0.2 compared with the observations. We note that this example comparison only considered annual results. In addition, in order to align with our benchmark dataset that has an annual time step, we used the December observations, which can be relatively easier given the less variability in the colder climate surrounding ABoVE sites in the winter season. In general, monthly dynamics at the global scale during all seasons will be more challenging and will be part of the future work. Finally, when comparing with in-situ observations, there is the common scale difference between the simulation data's resolution and the site's coverage, which may lead to larger differences in regions with higher heterogeneity (the differences are not necessarily model errors). For more validation information, please refer to [38, 40, 22].

**Coarse-resolution ED baseline at 2°×2° resolutions.** To better understand the trade-off between computational efficiency and quality, we constructed a coarse-resolution baseline by aggregating ED outputs to 2°×2° cells to represent the case where it takes roughly 1/16 of time to run the process-based model. This helps evaluate if it is important to run faster emulators at higher resolution, or if one can simply run process-based simulation at much coarser resolution without reducing the quality by too much (i.e., limited heterogeneity in data). The errors as a result of the downsampling are then calculated as the differences to the original 0.5°×0.5° data. The results show that the coarse-resolution ED baseline produced significantly higher errors across all target variables, such as 4.778 RMSE for height and 2.030 RMSE for AGB, leading to 151.5% and 313.4% error increase compared with the best-performing emulators. The degraded performance highlights the importance of emulators for high-resolution modeling to capture spatial heterogeneity and fine-scale ecological dynamics, further underscoring the advantages of emulators for computational cost reduction.

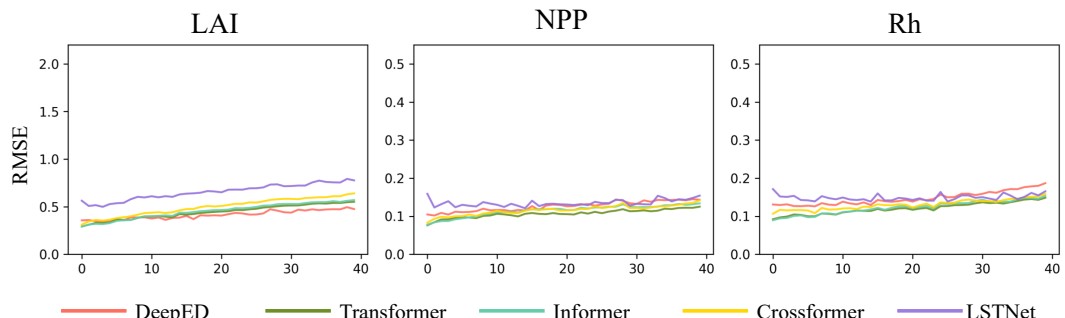

Figure 10: Visualization along the temporal dimension: RMSE over 40 years.

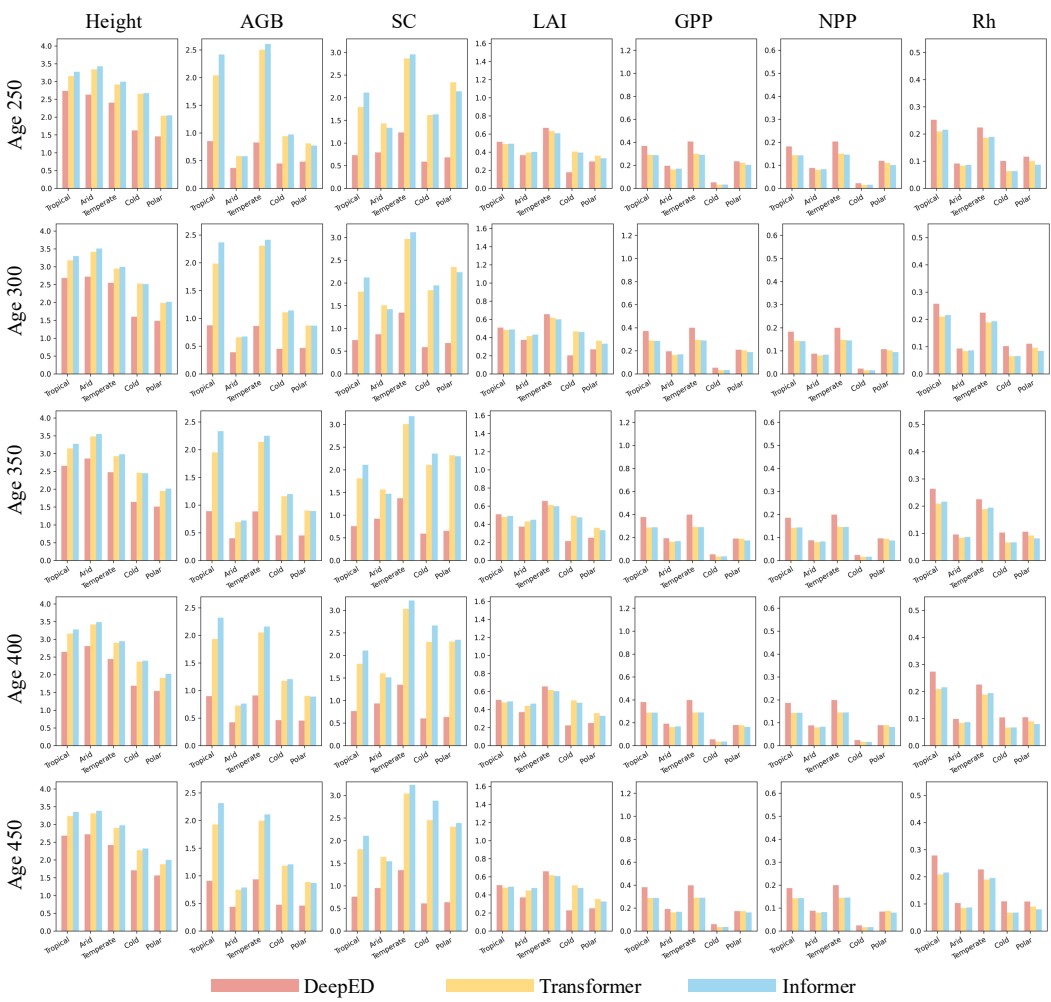

Figure 11: Error distributions among different climate zones and forest ages (Part 1).

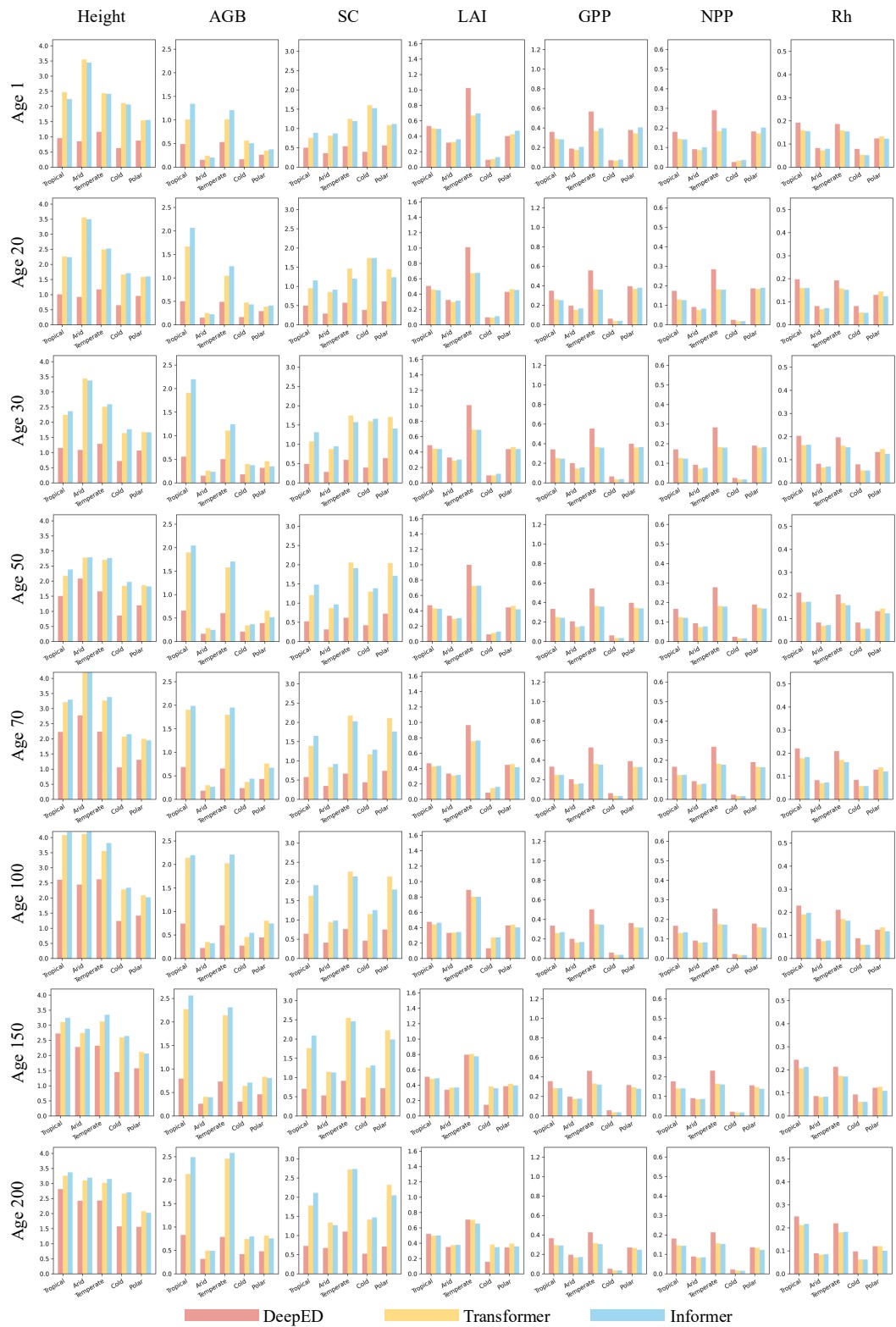

Figure 12: Error distributions among different climate zones and forest ages (Part 2).

