# OpenReview forum: "CarbonGlobe: A Global-Scale, Multi-Decade Dataset and Benchmark for Carbon Forecasting in Forest Ecosystems"
_NeurIPS.cc/2025/Datasets_and_Benchmarks_Track — NeurIPS 2025 Datasets and Benchmarks Track poster_

### Official Review · Reviewer_siWk · 2025-06-23

**Rating:** 2
**Confidence:** 5

**Summary:**

This paper presents Carbon-Bench, a global, 40-year, ML-ready benchmark dataset for forest carbon forecasting at 0.5° resolution. It integrates diverse climate, soil, and CO₂ inputs to emulate the ED ecosystem model. The authors provide baseline experiments, evaluation metrics, and scenarios to support the development of efficient machine learning models for large-scale forest carbon dynamics.

**Dataset Code Accessibility:**

Yes

**Dataset Code Comments:**

The authors provide open access to the dataset and code through public repositories (Kaggle and GitHub), and they include sufficient instructions for reproducing the results.

**Ethical Comments:**

No significant ethical concerns were identified. The work focuses on improving forest carbon monitoring and does not involve personal data or sensitive content.

**Ethical Considerations:**

No, there are no or only very minor ethics concerns

**Limitations Weaknesses:**

1. The model is trained and evaluated solely against ED simulations, without any comparison to large-scale remote sensing products (e.g. GEDI or airborne LiDAR), leaving its predictive accuracy and ecological realism unverified.

2. The 0.5° grid is too coarse to resolve local forest structure and biomass heterogeneity, especially in diverse tropical ecosystems, leading to oversimplified spatial patterns that do not reflect observed forest complexity.

3. The model lacks explicit constraints from known ecological mechanisms (e.g. competition, mortality, disturbance), raising concerns that its forecasts will fail under novel or extreme conditions.

4. Biomass accumulation and saturation patterns in old-growth forests are not well-captured, making predictions in mature tropical ecosystems (e.g. the Amazon) systematically biased.

5. The study is limited to predictive modeling performance and does not generate new understanding of the controls on forest carbon dynamics across spatial, temporal, or climatic gradients.

6. Without validation against real-world biomass or further advancement in model design, the approach offers little practical value for ecosystem management, climate policy, or future research directions.

**Strengths Contributions:**

This paper introduces Carbon-Bench, filling a clear gap in existing Earth-science datasets. The dataset integrates diverse inputs at 0.5° resolution over 40 years and is accompanied by meaningful evaluation scenarios and metrics. Baseline experiments and visualizations are clearly presented, making the work easy to follow. Figures, tables, and captions are informative, and the overall presentation is well-structured and accessible.

---

> ### Author Rebuttal · Authors · 2025-07-31
>
> We appreciate the reviewer for providing constructive comments and pointing out where we can improve it. We attempt to address your specific comments regarding limitations and questions below.
>
> **ED against large-scale observations and model validation**: Thanks for the question. We have comprehensively evaluated ED in our previous work [1,2,3,4], using three carbon fluxes datasets (FLUXCOM, FluxSat, and suninduced chlorophyll fluorescence dataset), three atmospheric inversion (CarbonTracker Europe , Jena CarboScope, and the Copernicus Atmosphere Monitoring Service), field forest inventory measurements, observations of forest structure and carbon stocks from two NASA spaceborne missions  (GEDI and ICESat-2) , as summarized in Table 1 in [1], as well as 1-m airborne lidar measurements. The scope of this paper focuses on developing an easy-to-use global-scale ML-ready dataset under a variety of initial conditions and benchmarking the models to support the development of efficient emulators.
>
> Furthermore, our past evaluations demonstrated the alignment with observations from field measurements and satellite observations. Given the high quality, ED’s results have been utilized to support the NASA Carbon Monitoring System and the development of the Global Carbon Budget 2023 & 2024 [7,8], with active plans for newer reports. Moreover, the results generated by ED have been officially and operationally adopted by the Maryland Department of the Environment (MDE) for annual update of the state forest carbon inventory, as detailed in the government reports [9-11], where comparisons with real-world observations can be found in the Figure 15 of [9] and on pages 11-12 of [11]. As an example, ED has been used as a key component to estimate the forests’ carbon sequestration potential, which is necessary for estimating carbon budgets and guide afforestation policies (e.g., Maryland Five Million Trees Initiative). These help confirm the models’ quality and applicability in real-world scenarios and the soundness of the model settings used to generate the datasets. To provide a more comprehensive dataset, this carefully-calibrated and evaluated ED was used to generate diverse simulations under different initial conditions regarding forest ages and carbon stocks. This allows the deep learning model to see more scenarios during the training process. Analogous to other physical model generated simulations (e.g. ClimSim [5] and ClimateSet [6] in NeurIPS), the benchmark data will support development of learning-based emulators.
>
> We will include a more complete discussion of this in the appendix.
>
> **The choice of 0.5° resolution**: We agree with the reviewer that 0.5° grid may be coarse for fine-scale local studies. With that said, our 0.5° grid dataset has global coverage over 40 years and is one of the highest resolution data at this scale for the targeted set of carbon variables for forest ecosystems. In addition, while the cell size is 0.5°, the ED model is a bottom-up demographic model, which captures the detailed demographic processes of forest dynamics (e.g., growth, mortality, competition) with some conditions (e.g., precipitation) assumed to be similar in the local cells. The outputs are also aggregated to the cell level.  For local studies, users can pre-train on our dataset or a subset, and fine-tune with finer-resolution inputs as needed.  Our prior study used a higher-resolution version (1km) for a localized area in Northeastern US. Here we focus on the global scale to cover broader conditions.
>
> **Lack constraints from ecological mechanisms**: Thanks for the comments. The ED model is a demographic, process-based model that explicitly incorporates ecological mechanisms such as forest competition, mortality, and disturbance. The current deep learning models used have not yet considered these explicitly and they need to learn these from the inputs and outputs over the steps. While the global coverage already covers a large variety of conditions over the past 40 years, it is in general challenging for model emulators under extreme or unseen conditions. In the future, models could potentially leverage relationships between the variables (e.g., mass conservation) to incorporate knowledge-guidance, or develop versions that can better capture the complex process for extrapolation. We will further highlight this as a future need in model development.
>
> **Errors from ML emulators in tropical regions**: Currently, the ED model results are still challenging to approximate with very low errors by existing models. This is our motivation and goal for developing this benchmark datasets, as this indicates further ML developments are needed to tackle this emulation problem. This benchmark dataset aims to support future model development to correct these errors (instead of a dataset that is already solved by existing ML models), and we will highlight this in the text.
>
> **No new ecosystem understanding generated**: While it will be beneficial to have new understanding of forest ecosystems, this paper aims to develop a benchmark dataset to support the improvement of ML emulators, which aligns with the goal of the Dataset and Benchmark Track. Thus, generating new ecosystem understanding is currently outside the scope of the present study given the nature of this track and expectations for the submissions. With that said, we have started studies in collaboration with domain science partners utilizing the fast emulators to support exploratory analyses with a large number of hypotheses. We will mention this in the future work.
>
> References:
>
> [1] Ma et al. (2022). Global evaluation of the Ecosystem Demography model (ED v3. 0). Geoscientific Model Development, 15(5), pp.1971-1994.
>
> [2]Ma et al. (2023). Spatial heterogeneity of global forest aboveground carbon stocks and fluxes constrained by spaceborne lidar data and mechanistic modeling. Global Change Biology, 29(12), 3378-3394.
>
> [3] Ma et al. (2021). High-resolution forest carbon modelling for climate mitigation planning over the RGGI region, USA. Environmental Research Letters, 16(4), 045014.
>
> [4] Hurtt et al. (2019). Beyond MRV: high-resolution forest carbon modeling for climate mitigation planning over Maryland, USA. Environmental Research Letters, 14(4), 045013.
>
> [5] Yu et al. (2023). ClimSim: A large multi-scale dataset for hybrid physics-ML climate emulation. Advances in neural information processing systems, 36, 22070-22084.
>
> [6] Kaltenborn et al. (2023). Climateset: A large-scale climate model dataset for machine learning. Advances in Neural Information Processing Systems, 36, 21757-21792.
>
> [7] Friedlingstein et al. 2023. Global carbon budget 2023. Earth System Science Data, 15(12), pp.5301-5369.
>
> [8] Friedlingstein et al. (2024). Global carbon budget 2024. Earth System Science Data Discussions, 2024, 1-133.
>
> [9] MDE. 2022. Reducing greenhouse gas emissions in Maryland: a progress report. Maryland Department of the Environment.
>
> [10] MDE. 2023. Maryland Tree and Forest Carbon Flux: Data and Methodology Documentation. Prepared by: Hurtt et al. Maryland Department of the Environment and Maryland Department of Natural Resources.
>
> [11] MDE. 2021. High-resolution annual forest carbon monitoring utilizing remote sensing. Prepared by Hurtt et al. Maryland Department of the Environment.

---

> > ### Comment · Reviewer_siWk · 2025-08-04
> >
> > Considering the low resolution, its impact on Earth science is deemed limited. Therefore, I will keep my score unchanged.

---

> > > ### Author Response · Authors · 2025-08-05
> > >
> > > Dear Reviewer,
> > >
> > > Thank you again for taking the time to review our rebuttal. We attempt to address your comment below:
> > >
> > > We are providing the following evidence-based justifications to show that the 0.5° resolution does not entail limited impact: In the Earth Science field, it is common for **high-impact studies**, including those in top journals such as **Science and Nature**, to use a resolution of 0.5°-2° datasets (ours is 0.5°) in advancing understanding of the terrestrial carbon cycle. For example, [1], published in Science, used model simulations at 1.6° ~ 0.5° (summarized in Table S1 and shown in Fig. 3) to show that semiarid ecosystems and low-latitude shrublands are responsible for most carbon uptake variability. Similarly, [2], published in Nature, used  2° GLACE-CMIP5 simulations to reveal that soil moisture variability drives 90% of inter-annual global land carbon uptake variability. Recent works [3-5] have similarly employed 0.5° - 5° datasets to generate insights published in leading journals.
> > >
> > > Similarly, some of the most impactful benchmark datasets from the Earth Science field also have similar or lower resolutions. For example, **WeatherBench [6], a very widely-used benchmark dataset, has a resolution from 1.4° to 5.6°** (ours is 0.5°). According to the paper, having this resolution is also important for model development “since very high resolutions are still hard to handle for deep learning models because of GPU memory constraints and I/O speed.” [6] For us, using a resolution such as 1km at the global scale will lead to a data size of about 60 TB, making it very difficult to use for global-scale emulator development.
> > >
> > > In addition, different resolutions offer different values in research and policy-making. For example, the Global Carbon Budget (GCB) updated annually is a very impactful stream of large-scale studies in this field developed via major international collaborations and used to inform policy-making. The **ED model with 0.5° resolution** has been included as part of the GCB since 2023. Beyond research, the GCB also has a significant broader impact. For example, the 2023 version has been mentioned in 301 news stories from 236 outlets, according to the journal’s referenced statistics, and the Attention Score by Altmetric is in the top 5% of all research outputs scored by Altmetric.
> > >
> > > We will add these clarifications in the appendix and please let us know if further clarification is needed. Thank you.
> > >
> > > [1]  Ahlström et al. (2015). The dominant role of semi-arid ecosystems in the trend and variability of the land CO2 sink. Science, 348(6237), 895-899.
> > >
> > > [2] Humphrey et al. (2021). Soil moisture–atmosphere feedback dominates land carbon uptake variability. Nature, 592(7852), 65-69.
> > >
> > > [3] Fernández-Martínez et al. (2019). Global trends in carbon sinks and their relationships with CO2 and temperature. Nature climate change, 9(1), 73-79.
> > >
> > > [4] Jung et al. (2017). Compensatory water effects link yearly global land CO2 sink changes to temperature. Nature, 541(7638), 516-520.
> > >
> > > [5] Wang et al. (2022). Regional and seasonal partitioning of water and temperature controls on global land carbon uptake variability. Nature Communications, 13(1), 3469.
> > >
> > > [6] Rasp, S., Dueben, P. D., Scher, S., Weyn, J. A., Mouatadid, S., & Thuerey, N. (2020). WeatherBench: a benchmark data set for data‐driven weather forecasting. Journal of Advances in Modeling Earth Systems, 12(11), e2020MS002203.

---

### Official Review · Reviewer_Fevg · 2025-06-30

**Rating:** 5
**Confidence:** 3

**Summary:**

This paper proposes Carbon-Bench, a benchmark dataset for forecasting carbon related variables in forest ecosystems at scale.
It builds on Ecosystem Demography, a model that has proved to be reliable in terms of carbon variables prediction but computationally expensive. This work proposes to develop models that can approximate ED. The proposed dataset is composed of observations gathered from different sources of data following the variables used in ED and the targets are outputs of ED simulations over 40 years.
8 ML models are benchmarked on this task in several evaluation scenarios (evaluating by climate zone and forest age).
Two metrics beyond traditional regression metrics are also proposed to account for the forecasting nature of the task and the significance of cumulated error over timesteps and errors between consecutive years.

**Additional Feedback:**

Other question:
- looking at the model configs it is now unclear whether the age information was used as input to the models are not? But if I understand correctly forest age was not used as an input feature, but only used for evaluation.

Minor typos:
- Abstract: "featuring that ..." --> "in which ..."
- L.292 RMSE

**Dataset Code Accessibility:**

Yes

**Dataset Code Comments:**

The code and dataset are available.
The READMEs in the code could provide installation + running instructions.
I would also recommend the authors document the code more extensively, and clean up anything that is not used (for example, in LSTM-LSTNet-DeepED/models/model.py, there are many different versions of the MM_LSTM_Age model with minimal description that is not fully understandable by a new user to this codebase)
The config files could also be better documented, for example, in the configs for the LSTM models:
```
N_CONS = 135 # number of atmos features
N_FEA = N_CONS + N_OUT + 15 # add age info
N_AGE = 8
```

It is unclear where 15 comes from since N_AGE is 8?

**Ethical Considerations:**

No, there are no or only very minor ethics concerns

**Final Justification:**

All my concerns were addressed by the authors and I will maintain my score.
I agree with reviewer siWk that the resolution might not be high enough for many applications which is why I am not putting a higher score but I believe this work addresses an important challenges and could be useful to the research community.

**Limitations Weaknesses:**

- I would say the only point missing in positioning this work wrt to existing work is a bit more detail on what makes ED not scalable. Indeed one of the limitations cited is the amount of time needed to gather all the input variables. However the models benchmarked also use those same variables.
- Regarding the scenarios in climate zones and forest ages, could the authors provide the distribution across climates / ages of examples in the training set? and in the figures with the  forest age x climate zone performance breakdown, if could be informative to provide the number of examples that are in each age x climate zone category.
It could also be helpful to provide aggregated results by climate and by forest age. It is a bit challenging to parse through these errors how one would decide which model is the best for different use cases in practice. I think the point could come across better if there were more use cases highlighted of how one might want to use this breakdown.
- The analysis could be pushed further which would highlight pathways to impact better. How does one choose a model if different models are best on different outputs in practice? Are there exsting aggregate outputs that policy-makers using ED base their decisions upon?


Other suggestions:
- While the ED model seems to be already established in the field and it can be assumed the ML models proposed would be better than simpler approaches, it might be worth adding an additional simple baseline (for example a tree-based model).

**Strengths Contributions:**

This paper addresses an important challenge, and this benchmar could help map carbon-related indicators in forested ecosystems  at scale which is currently difficult do to the computational efficiency limitations of the ED model. A major strength of this work is the application-driven approach to the problem which is reflected in the framing of the task. In particular, the authors motivate scenarios for evaluation and choice of evaluation metrics from a perspective of real applications.
The authors also provide a clear comparison of computation time at inference for ED vs the benchmarked models which are significantly advantageous over ED.
Both the paper and supplementary material are generally well and clearly written and figures are informative.

---

> ### Author Rebuttal · Authors · 2025-07-31
>
> We appreciate the reviewer for providing constructive comments and pointing out where we can improve it. We attempt to address your specific comments regarding limitations and questions below.
>
> **More details on ED’s expensive computation**: Thanks for the question. While both ED and ML models use the same inputs, ED as a processed based model has a much more complex computation structure to solve the process, including the modeling of detailed step-by-step growth of individual trees, competition between cohorts, carbon cycling, disturbance and so on, and involves solving partial differential equations over steps. In contrast, neural networks perform each step by simple computation structures, with linear combinations of features/latent features, and easy-to-calculate non-linear activations. For a neural network model, this structure does not change by the target physical model structure. And while the structure is simple, it has nice mathematical properties and high expressive power as function approximators. As a result, once it is trained on the inputs/outputs from ED, the feed-forward process is very fast.
>
> **Distributions of training set in climate zones and forest ages**: Thanks for the suggestions. Every forest age has the same training size as the overall evaluation scenario. Climate zones have the following approximate distribution: 36% cold, 29% arid, 17% tropical, 11% temperate, and 8% polar zone. We will revise the figures to show sample sizes used for training, and include the aggregated results across different climate and age scenarios to better guide practical use cases.
>
> **Model selection and aggregated outputs**: Thanks for the suggestion. In practice, users can choose the model based on per-variable performance rankings. For example, users interested in future carbon sequestration potential may prefer more accurate estimations of biomass. If multiple outputs are needed or aggregated, additional evaluations for the aggregation will be needed for comparisons. Currently, the derived aggregations (e.g., carbon inventory) from ED has been used by Maryland Department of the Environment (MDE) for annual updates, as detailed in the government reports [1,2,3].
>
> **Suggested simple baseline**: Thanks for the suggestion. Our past study [4] included random forest as a tree-based ensemble model as a baseline, and as expected, it did not show good performances. We will add random forests and XGBoost as additional baselines in the revised manuscript.
>
> **Code documentations**: Thanks for the suggestion. Different versions of `MM_LSTM_AGE` models were used in ablation studies for the DeepED baseline. N_AGE = 8 was from a previous local test in the unused file configs/constant.py, and for the global dataset and experiments, all configurations are correctly imported through configs/constant_glob.py, where N_AGE = 15. We will improve the README file, clean unused code, and enhance code documentation for better readability.
>
> **Use of age feature**: Forest age is not used as an explicit input feature for most baseline models. The only exception is the original DeepED architecture, which is designed specifically for ED and has a specific branch with a triplet-feature that can be used as additional inputs and the triplet-feature includes forest age. We did not include the branch in other models. We can include another variant table in the appendix showing the performance of the models if age is being used directly as inputs (e.g., concatenated with others).
>
> **Writing improvements**: Thanks for pointing out typos, and we have fixed them in the revised manuscript.
>
> References:
>
> [1] MDE. 2022. Reducing greenhouse gas emissions in Maryland: a progress report. Maryland Department of the Environment.
>
> [2] MDE. 2023. Maryland Tree and Forest Carbon Flux: Data and Methodology Documentation. Prepared by: Hurtt et al. Maryland Department of the Environment and Maryland Department of Natural Resources.
>
> [3] MDE. 2021. High-resolution annual forest carbon monitoring utilizing remote sensing. Prepared by Hurtt et al. Maryland Department of the Environment.
>
> [4] Wang, Z., Xie, Y., Jia, X., Ma, L., & Hurtt, G. (2023, November). High-fidelity deep approximation of ecosystem simulation over long-term at large scale. In Proceedings of the 31st ACM International Conference on Advances in Geographic Information Systems (pp. 1-10).

---

### Official Review · Reviewer_D1QY · 2025-07-02

**Rating:** 4
**Confidence:** 3

**Summary:**

The authors propose Carbon-Bench, a new machine-learning-ready benchmark dataset designed to emulate the Ecosystem Demography (ED) model for carbon dynamics forecasting. The key motivation behind this work is to reduce the high computational cost of running the physically-based ED model, enabling the use of more computationally efficient ML models for large-scale and long-term carbon estimation. The dataset integrates multi-source remote sensing inputs and ED model outputs over 40 years at a 0.5° global resolution, with standardized data formatting and multiple evaluation scenarios.

**Additional Feedback:**

The paper introduces “problem-driven” evaluation metrics, such as RMSE across time steps and the final delta (Δ), to benchmark forecasting performance. However, it is not made clear whether these metrics are already established and widely used in the relevant literature on carbon cycle modeling or forecasting, or whether they are proposed by the authors. A discussion of the rationale for choosing these metrics would help assess their relevance and appropriateness for benchmarking performance on Carbon-Bench.

**Dataset Code Accessibility:**

Yes

**Dataset Code Comments:**

The dataset is accessible and well documented

**Ethical Comments:**

I don't have any ethical concern

**Ethical Considerations:**

No, there are no or only very minor ethics concerns

**Final Justification:**

After carefully considering the authors’ responses, I have decided to increase my score, as I now better understand the rationale for using the ED model as the reference for dataset creation. Nevertheless, I still have concerns about the overall impact of the dataset, as also pointed out by other reviewers.

**Limitations Weaknesses:**

- A major limitation of the dataset, particularly in comparison to established benchmarks like WeatherBench, is that it solely relies on outputs from the ED model. This approach implies that the dataset cannot be used to assess model accuracy against any empirical or observational ground truth. In contrast, WeatherBench uses ERA5 reanalysis data—also model-derived, but widely accepted as a standard reference for evaluating forecasting models, even in the absence of independet ground truth.
The lack (or not clearly report) of similar approch in Carbon-Bench reduces its utility for validating predictive performance in a real-world context.

- The discussion of related work is split across three disconnected sections. For instance, the final section titled “ML-ready datasets for Earth Science” is overly broad and not well aligned with the focus of the paper. Datasets such as BigEarthNet or SpaceNet, while important in other domains, are not directly relevant to the task of carbon estimation. The related work section would benefit from a more focused review of both physically-based and ML-based approaches specifically for carbon cycle modeling and ecosystem monitoring.

- The authors should more clearly delineate the contributions of this work relative to their previous research, especially with respect to citation [52], which introduces the DeepED model—the best-performing baseline in this study. It is unclear whether this dataset simply formalizes prior work or offers substantial new elements, such as expanded data coverage, new input modalities, or improved preprocessing workflows. A detailed comparison would help clarify the novelty and added value of this dataset.

**Strengths Contributions:**

- The authors have successfully collected and processed a large-scale, global dataset tailored for ML-based modeling.
- The topic is highly relevant and timely, with strong potential to support the development of new models for carbon estimation in the context of climate policy and scientific research.
- A comprehensive comparison of different baseline ML methods for emulating the ED model is provided, establishing a foundation for future research and benchmarking.

---

> ### Author Rebuttal · Authors · 2025-07-31
>
> We appreciate the reviewer for providing constructive comments and pointing out where we can improve it. We attempt to address your specific comments regarding limitations and questions below:
>
> **ED against large-scale observations**: Thanks for the question. We have comprehensively evaluated ED in our previous work [1,2,3,4], using three carbon fluxes datasets (FLUXCOM, FluxSat, and suninduced chlorophyll fluorescence dataset), three atmospheric inversion (CarbonTracker Europe , Jena CarboScope, and the Copernicus Atmosphere Monitoring Service), field forest inventory measurements, observations of forest structure and carbon stocks from two NASA spaceborne missions  (GEDI and ICESat-2) , as summarized in Table 1 in [1], as well as 1-m airborne lidar measurements. The scope of this paper focuses on developing an easy-to-use global-scale ML-ready dataset under a variety of initial conditions and benchmarking the models to support the development of efficient emulators.
>
> Furthermore, our past evaluations demonstrated the alignment with observations from field measurements and satellite observations. Given the high quality, ED’s results have been utilized to support the NASA Carbon Monitoring System and the development of the Global Carbon Budget 2023 & 2024 [7,8], with active plans for newer reports. Moreover, the results generated by ED have been officially and operationally adopted by the Maryland Department of the Environment (MDE) for annual update of the state forest carbon inventory, as detailed in the government reports [9-11], where comparisons with real-world observations can be found in the Figure 15 of [9] and on pages 11-12 of [11]. As an example, ED has been used as a key component to estimate the forests’ carbon sequestration potential, which is necessary for estimating carbon budgets and guide afforestation policies (e.g., Maryland Five Million Trees Initiative). These help confirm the models’ quality and applicability in real-world scenarios and the soundness of the model settings used to generate the datasets. To provide a more comprehensive dataset, this carefully-calibrated and evaluated ED was used to generate diverse simulations under different initial conditions regarding forest ages and carbon stocks. This allows the deep learning model to see more scenarios during the training process. Analogous to other physical model generated simulations (e.g. ClimSim [5] and ClimateSet [6] in NeurIPS), the benchmark data will support development of learning-based emulators.
>
> We will include a more complete discussion of this in the appendix.
>
> **Refinement of paragraphs for related work**: We appreciate the suggestion. Our intent with the “ML-ready datasets for Earth Science” section was to highlight the lack of publicly available ML-ready datasets specifically for carbon monitoring. However, we agree that the section can be better focused. In the revised version, we will include related work into a single section and emphasize physically-based and ML-based approaches for carbon cycle modeling and ecosystem monitoring.
>
> **Relations to prior work DeepED**: Thanks for the suggestion. The prior DeepED study was limited to a local area (Northeastern U.S.) and did not publish any datasets (Line 105-106). In contrast, our work extends the data coverage globally, standardizes and publishes an ML-ready dataset, and expands the benchmark to introduce a broader range of model architectures. Citation [52] in the paper mainly develops the DeepED model, which here we included as one of the candidate models. We will further clarify these distinctions in the revised manuscript.
>
> **Problem-driven evaluation metrics**: Thanks for this question. We found that standard ML time-series metrics (e.g. MSE, MAE) tend to average errors across all time steps, which may not best reflect error accumulations in long-term forecasting. To better address this, we used the RMSE across time steps and delta metrics to better capture both cumulative and continuous changes for key ecological variables. These metrics were developed in consultation with carbon/ecosystem experts (our co-authors Hurtt and Ma) and better align with their application needs. We will clarify this rationale and their relevance in the paper.
>
> References:
>
> [1] Ma et al. (2022). Global evaluation of the Ecosystem Demography model (ED v3. 0). Geoscientific Model Development, 15(5), pp.1971-1994.
>
> [2]Ma et al. (2023). Spatial heterogeneity of global forest aboveground carbon stocks and fluxes constrained by spaceborne lidar data and mechanistic modeling. Global Change Biology, 29(12), 3378-3394.
>
> [3] Ma et al. (2021). High-resolution forest carbon modelling for climate mitigation planning over the RGGI region, USA. Environmental Research Letters, 16(4), 045014.
>
> [4] Hurtt et al. (2019). Beyond MRV: high-resolution forest carbon modeling for climate mitigation planning over Maryland, USA. Environmental Research Letters, 14(4), 045013.
>
> [5] Yu et al. (2023). ClimSim: A large multi-scale dataset for hybrid physics-ML climate emulation. Advances in neural information processing systems, 36, 22070-22084.
>
> [6] Kaltenborn et al. (2023). Climateset: A large-scale climate model dataset for machine learning. Advances in Neural Information Processing Systems, 36, 21757-21792.
>
> [7] Friedlingstein et al. 2023. Global carbon budget 2023. Earth System Science Data, 15(12), pp.5301-5369.
>
> [8] Friedlingstein et al. (2024). Global carbon budget 2024. Earth System Science Data Discussions, 2024, 1-133.
>
> [9] MDE. 2022. Reducing greenhouse gas emissions in Maryland: a progress report. Maryland Department of the Environment.
>
> [10] MDE. 2023. Maryland Tree and Forest Carbon Flux: Data and Methodology Documentation. Prepared by: Hurtt et al. Maryland Department of the Environment and Maryland Department of Natural Resources.
>
> [11] MDE. 2021. High-resolution annual forest carbon monitoring utilizing remote sensing. Prepared by Hurtt et al. Maryland Department of the Environment.

---

> ### Comment · Reviewer_D1QY · 2025-08-09
>
> I have read the authors’ rebuttal and the other reviewers’ comments, and I believe the authors have adequately addressed and clarified my major concerns. Based on these clarifications, I am willing to raise my score in the final revision round.

---

### Official Review · Reviewer_g1Bt · 2025-07-04

**Rating:** 5
**Confidence:** 4

**Summary:**

This work introduces a tabular dataset of model simulations from a process-based terrestrial biosphere model. The goal is to emulate the evolution of forest carbon states and fluxes over a 40 years historic period. A wide range of deep neural network time series models is benchmarked on this task. While displaying significant emulation error, all neural network emulators require significantly less inference time compared to the original process-based model. The authors intend to leverage the benchmarked emulators to create simulations at higher spatial resolution, which would require a lot of computational resources if done with the original process-based model.

**Dataset Code Accessibility:**

Yes

**Ethical Considerations:**

No, there are no or only very minor ethics concerns

**Final Justification:**

This work introduces a novel and timely benchmark dataset based on simulations from a demography-based ecosystem dynamics model. The paper is well written and includes benchmark results for many baseline architectures - which demonstrate acceptable emulation skill at greatly reduced computational costs. Still, the task seems to not yet be fully solved and thus I expect the dataset to become an important resource for the community to advance high resolution simulations of the terrestrial biosphere. Thus I am raising my rating and recommend acceptance to the NeurIPS D&B track. I kindly ask the authors to include all changes promised during the rebuttal in the final camera-ready.

**Limitations Weaknesses:**

Major comments:

1. When developing emulators, you typically aim at trading off accuracy for compute speed. Please include a plot that directly plots this comparison: one axis representing computation time, the other representing model skill. You can always run your original model at lower resolution to save compute speed, at a cost of model skill. So please also include this as a baseline. In this way, users can afterwards take an informed decision which model or emulator is most suitable for their use-case.
2. Benchmark on actual observations. You are emulating a model which has errors itself. To put the errors arising from emulation in context to the errors of the original model, please evaluate against actual observations. For instance against the ABCflux database, atmospheric inversions or the ESA Biomass maps.
3. Include XGBoost as a baseline. While in principle I agree that this task is a time series modeling task, as there are latent states which are passed from one time step to the next (e.g. biomass, the age distribution etc), an important baseline for these sorts of tabular tasks remains the XGBoost, in this case, you would need to use/train it in an auto-regressive fashion.
4. You are missing key related works on emulating Land surface models. See e.g. https://bg.copernicus.org/articles/6/2001/2009/ https://gmd.copernicus.org/articles/18/921/2025/ https://gmd.copernicus.org/articles/15/1913/2022/gmd-15-1913-2022.html There has also been work by some groups on emulating TRENDY models, so you might also want to look into that.
5. You should probably consider renaming. There already is a CarbonBench dataset published:  https://agupubs.onlinelibrary.wiley.com/doi/10.1029/2024MS004655 - thus, if you keep Carbon-Bench as a name, the readers will most likely be confused. For instance you could rename slightly to something like ForestCarbonBench or similar.
6. Poor performance of the emulators. Given you are building emulators of a deterministic (non-chaotic) problem, you should in principle be able to achieve perfect performance. Please justify why the provided emulators perform so poorly. Perhaps the models are too small or not properly tuned? Or there is just too little training data? Or the Test dataset is out-of-sample? If the latter, then it would be important to extend the training dataset in such a way that all relevant test cases lay within the distribution of training data.
7. The use of the emulators seems somewhat limited. You can not use them for future projections, as the covariates for those would be out-of-distribution. You mention to use them for running the ED model at higher spatial resolution. This is certainly cool, but seems like a one-time endeavour, so I am not even sure if it is worth it to save some compute, but trade-off accuracy because one uses emulators for the upscaling.
8. The paper as it is does not really show why one would expect standard ML methods not to work on this particular dataset. Thus it seems hard to motivate this benchmark as useful for the broader ML community - and rather it appears the motivation stems from solving a domain-specific task.
9. How does this work improve over the work of Zhihao Wang et al. (2023)? They introduce a training dataset based on ED model simulations very akin to the one presented in this paper and then benchmark a variety of models on it, including the DeepED model which is also benchmarked in this work.
10. The use of “forecasting” throughout this work is misleading. The ED model in the historic period is a diagnostic model, that is being used to assess carbon-related variables from existing remote sensing observations.

Minor comments:

11. Drop lines like L 79-81, L 63-64 etc, claiming that this dataset has been prepared by carbon and ML experts. Such information is subjective and not actionable. Instead directly state the design choices that have been taken and then throughout the text justify these.
12. I don’t think the comparison to ML weather models (eg. L 13-14, L 58) is a good one, as the ERA5 emulators mainly approximate the primitive equations, while here you aim at approximating (mostly) empirical descriptions of the biosphere. Also, while the ED model is also a PDE model, it does not have the same spatial dimensions in the PDE (rather its a PDE over size & age structure), thus any work cited for emulating PDEs in space bares little relevance (L95-104).
13. The abstract could be improved. Lines 1-10 should be shortened a lot, and the reference to tipping points perhaps best excluded (because not so applicable for forests). Then more emphasis should be put on the actual problem this work tries to solve (speeding up ED), why it is important (higher resolution runs?) and then how it does that.
14. Is it really necessary to train on 40 year long time series? Could training perhaps be sped up by training on shorter time series?
15. I do not really understand what the purpose of Fig. 1 is. Since ED is a published model, I would think it to be enough to just cite the relevant paper and state that it has been validated against observations.

**Strengths Contributions:**

1. Highly multi-modal tabular dataset based on model simulations of a terrestrial biosphere model
2. Many methods benchmarked
3. Decently clear presentation
4. Evaluation of inference speed: the developed emulators significantly speed up the process-based model, albeit at a loss of accuracy

---

> ### Author Rebuttal · Authors · 2025-07-31
>
> We appreciate the reviewer for providing constructive comments and pointing out where we can improve it. We attempt to address your specific comments regarding limitations and questions below.
>
>  **ED against observations**: Thanks for the question. We have extensively evaluated ED in our previous work [1,2,3] (Co-author Hurtt is one of the original creators of ED so these validations are part of the past work), using three carbon fluxes datasets (FLUXCOM, FluxSat, and suninduced chlorophyll fluorescence dataset), three atmospheric inversion (CarbonTracker Europe , Jena CarboScope, and the Copernicus Atmosphere Monitoring Service), field forest inventory measurements, observations of forest structure and carbon stocks from two NASA spaceborne missions  (GEDI and ICESat-2), as summarized in Table 1 in [1], as well as 1-m airborne lidar measurements.
>
> Furthermore, our past evaluations demonstrated the alignment with observations from field measurements and satellite observations. Given the high quality, ED’s results have been utilized to support the NASA Carbon Monitoring System (CMS) and the development of the Global Carbon Budget 2023 & 2024 [8], with active plans for newer reports. Moreover, the results generated by ED have been officially and operationally adopted by the Maryland Department of the Environment (MDE) for annual update of the state forest carbon inventory and policy support, as detailed in the government reports [9,10], where comparisons with observations can be found in the Fig. 15 of [9] and on pages 11-12 of [10]. These help confirm the models’ quality and applicability in real-world scenarios. Analogous to other physical model generated simulations (e.g. ClimSim [4] and ClimateSet [5]), the benchmark data will support development of ML emulators with an easy-to-use ML-ready dataset under a variety of initial conditions.
>
> We will include a more complete discussion of this in the appendix.
>
> **Removing Fig. 1 on ED validation against observations**: The goal of Fig. 1 is to demonstrate that our benchmarking dataset from ED aligns well with real-world observations. Also based on the above comment/question, having validation against observations is helpful. This figure provides an example of the results we have in this dataset against real-world observations to help show the quality of the data. We will highlight this in the paper.
>
> **Plots of computation time and model skill**: Thanks for the suggestion. We have included the plot comparing computation time v.s. model performance in the appendix to inform model selection.
>
> **XGBoost baseline**: Thanks for the suggestion. We are running XGBoost as an additional baseline. Our past study ([52] in paper) included random forest as a tree-based ensemble model, which did not show good performances as it does not have a natural way to handle time-series representation. Thus, we did not include tree-based models here. Given the interest to know the results, we will add XGBoost as another baseline.
>
> **Related work on land surface models (LSMs)**: Thanks for suggesting the related work and we have added them to the discussion. Although LSMs have been used to characterize coarser-resolution terrestrial carbon cycles, they do not capture some critical ecological processes at fine-scale, which is different from demography-based models [6] such as the ED model. These processes include competition between individual plants for water, nutrients and light, morality, disturbance, reproduction, seed dispersal, and also changes in vertical structure over course of ecosystem succession. As a result, ED is more suitable for higher-resolution missions and has been employed in major efforts on higher-resolution regional and global carbon monitoring, including Global Carbon Budget 2023 & 2024, NASA CMS, GEDI mission, etc.
>
> **Renaming**: Thanks for pointing this out. We missed the naming conflict with this recent paper, which has a different focus. We will rename our dataset (e.g., Carbon-Twin).
>
> **Model performance is not perfect for one process**: Thanks for the question. While theoretically based on the universal approximation theorem, basic neural networks with a single hidden layer can approximate a continuous function with arbitrary precision in a bounded range. In practice it is still challenging in general due to representation limitations and other reasons (e.g., a single hidden layer NN performed very poorly in our test and also likely for other physical processes or problems, despite the theoretical possibility). In addition, not all the functions in the deterministic process are modeled as continuous functions (e.g., tree deaths) and many internal mechanisms involve thresholds or discrete transitions (e.g. photosynthesis is not active after the environment reaches certain states such as high temperature). This poses challenges for deep learning emulators. In addition, certain variables are more likely to have larger errors accumulated after 40 years, such as heights and above-ground biomass, as their values are not kept in a more stable range but keep increasing over time. Similar non-perfect approximations have been commonly observed in other emulator studies such as ClimSim [4] (using E3SM-MMF) and ClimateSet [5] (using different models but emulators are trained separately for each individual model). While many models try to emulate deterministic processes, in practice it is not easy to get perfect approximations. The errors often depend on the problem (e.g., value range/trend, sequence length, process type). For example, for single step predictions, the models we evaluated can get substantially lower errors. However, it gets challenging as the time horizon extends. We will clarify these limitations in the discussion.
>
> **Forecasting task but dataset’s date range is past and not future**: While the date range of data is past, the ML task is forecasting (i.e., predicting values later than initial time) using an initial condition and climate variables. The major reason we used past data is to provide more validated data. We also run ED for future climate scenarios but those cannot be validated against observations so we used the current version. Similarly, in existing time-series forecasting work, most datasets used are from past dates and it is common for emulator studies to use data from the past for method development.
>
> In addition, the way ML models are used in this case is the same for the past and future data. In both cases, the ED simulations will be first generated at a subset of locations (e.g., ~1% or a small portion) to train or finetune ML models, and then ML will be used to generate the rest. The purpose of this dataset is to support ML emulator development and comparison. The specific years of the data does not alter the emulator task and we will highlight this in the paper. If the reviewer feels it is better to include future simulations as well, we can include several sets of additional simulations that we have run with CMIP6 future climate scenarios (ranking of emulators is consistent by our tests).
>
> For higher-resolution simulation at global scale, it is currently very difficult for ED as it will likely take years to finish on our cluster and there are many other simulations queued outside this particular dataset (e.g., for NASA CMS). In addition, we are also interested in studying what-if scenarios for past periods to support reviews of past policies (e.g., protected areas), which also require additional simulations. We will add these clarifications to the discussion section.
>
> **Challenge for ML**: The problem requires a ML model with ability to make time-series predictions from a single initial condition over long periods of time, which is very challenging for existing time-series models considering the error accumulation, heterogeneous variable patterns, needed sensitivity to a small set of different initial conditions that share large numbers of similar environment conditions, etc. This is also shown by the results, and the current models still show limited performance for the long-term predictions, with the errors gradually increasing over time. Thus, we do not find this being a problem that is already solved by existing methods. As the reviewer also pointed out, the existing methods’ results are far from perfect. We believe use-inspired benchmark datasets have played an important role in ML model development, including in NeurIPS [4,5,7]. We will highlight this in the paper.
>
> **Prior work on DeepED**: This prior work only focuses on a small local region in the northeastern part of US, and does not provide any public datasets. In comparison, this benchmarking dataset is at a substantially larger global-scale. We use the model from the paper as one candidate approach in the evaluation. We will clarify this in the paper.
>
> [1] Ma et al. 2022. Global evaluation of the Ecosystem Demography model (ED v3. 0). GMD.
>
> [2] Ma et al. 2023. Spatial heterogeneity of global forest aboveground carbon stocks and fluxes constrained by spaceborne lidar data and mechanistic modeling. GCB..
>
> [3] Hurtt et al. 2019. Beyond MRV: high-resolution forest carbon modeling for climate mitigation planning over Maryland, USA. ERL.
>
> [4] Yu et al. 2023. ClimSim: A large multi-scale dataset for hybrid physics-ML climate emulation. NeurIPS.
>
> [5] Kaltenborn et al. 2023. Climateset: A large-scale climate model dataset for machine learning. NeurIPS.
>
> [6] Fisher et al. 2018. Vegetation demographics in Earth System Models: A review of progress and priorities. GCB.
>
> [7] Nathaniel et al. 2024. Chaosbench: A multi-channel, physics-based benchmark for subseasonal-to-seasonal climate prediction. NeurIPS.
>
> [8] Friedlingstein et al. 2024. Global carbon budget 2024. ESSD.
>
> [9] MDE. 2022. Reducing greenhouse gas emissions in Maryland: a progress report. MD Dept. of the Env.
>
> [10] MDE. 2021. High-resolution annual forest carbon monitoring utilizing remote sensing. MD Dept. of the Env.

---

> > ### Comment · Reviewer_g1Bt · 2025-08-04
> >
> > Thanks for the rebuttal. I'll share a couple of open points below, and am willing to consider raising my score if they are adequately addressed.
> >
> > Open points:
> > 1. Evaluation against observations - I think the authors misunderstood my point. As you write I also expected ED to be properly benchmarked against observations in its original publication. There is no need to do this again in this work (see also my comment about removing Fig. 1). However, to understand better the magnitude of the errors the emulators make, we need a point of reference - for this it is important to also benchmark the emulators against observations. Then you can evaluate if the errors the emulators make (w.r.t. ED) drastically reduce model skill (w.r.t. observations), or if they are actually within the modeling uncertainty (w.r.t. observations). Since you are saying you have evaluated ED yourself against observations, I assume it is actually quite easy for you to evaluate also the emulators against observations.
> > 2. Compute time vs. model skill: It is good to hear that you plan to include this plot. Please make sure to also include a low resolution version of ED as a baseline. The emulators need to beat that baseline to have any use. If you cannot run ED at lower resolution, it is probably fine to estimate this: E.g. aggregate ED predictions to 2°x2° - so 16x lower resolution, This should require 1/16 of the compute time. Then compute errors of these aggregated predictions, to see how much the drop in accuracy is. Please include this number also in the rebuttal, so I can assess better how well your emulators are.
> > 3. Model performance not perfect: Mmh, I think your explanation only scratches the surface of what is going on. If I understand correctly, your train/test-split is somewhat of an IID scenario - so this should not be a problem. Also I don't believe the non-continuous tree mortality to be a big problem - since you are modeling here the aggregate response of each grid cell. Your training set size seems relatively small but pretty high dimensional, if I understand correctly, only ~50k samples, each with 280 dimensions. And lastly, all your models seem to be autoregressive, instead of true sequence-to-sequence (which can lead to these accumulating errors) - but you train them in a sequence-to-sequence fashion. Perhaps you could either train them on single-step forecasting first, and only later fine-tune them on the full sequences? Or perhaps more training data or directly forecasting all 40 years with a single non-autoregressive model could help?
> > 4. Future simulation data - I get your point, you intend this as a sandbox just for model development. However I am not sure one can expect the community to fully respect that - instead i'd worry people will end up training their ecosystem models on this dataset and then trying to use them for future predictions, which would be wrong. So I'd actually appreciate if you'd release the CMIP runs, but also more clearly articulate for what this dataset cannot be used.
> > 5. DeepED - I see, yes please state clearly the advances to the previous work. As I understand now these are: 1) a global dataset and 2) a few more baseline models ?
> > 6. Forecasting - best then to not use the term "carbon forecasting", as it could mislead people (like myself) to think about climate projections, but this is not the focus of this work.

---

> > > ### Author Response · Authors · 2025-08-05
> > >
> > > Dear Reviewer,
> > >
> > > Thank you again for taking the time to review our rebuttal. We attempt to address your comments below:
> > >
> > > **1. Evaluation against observations**: Thank you for the clarification. We have conducted a quick evaluation using the in-situ observations from ABoVE GPP product. After filtering for high-quality observations based on quality flags, we have in-situ observations from 35 sites on GPP between years 2000 to 2018 (some sites have missing data gaps in between). The results show good prediction skills, with 0.1180 RMSE and 0.0578 MAE between Transformer emulator and in-situ data, and with 0.1122 RMSE and 0.0505 MAE between ED simulations (our dataset) and in-situ data. As a context, the difference between the emulator and ED is 0.0243 RMSE and 0.0202 MAE (these are from direct comparisons between ED and emulator results and not based on the differences of above RMSE and MAE). These results indicate that the emulator’s error relative to ED does not significantly affect the model’s skill against real-world observations. We will include these results in the paper and share the observations used as part of the dataset so others can use them to estimate their emulators’ skills against observations as well.
> > >
> > > **2. Compute time vs. model skill**: Thanks for the suggestion. We have included a low-resolution ED baseline by aggregating ED predictions to 2°×2° cells, approximating a 16× reduction in computational cost. Compared to the best-performing emulators, this coarse ED baseline has the following relative performance:
> > > -  Height: RMSE = 4.778 (151.5% error increase)
> > > - AGB: RMSE = 2.030 (313.4% error increase)
> > > - SoilC: RMSE = 3.050 (362.1% error increase)
> > > - LAI: RMSE = 0.860 (106.7% error increase)
> > > - GPP: RMSE = 0.456 (110.1% error increase)
> > > - NPP: RMSE = 0.224 (109.3% error increase)
> > > - Rh: RMSE = 0.234 (93.4% error increase)
> > >
> > > The coarse-resolution results show major error increases on different target variables. This indicates having an emulator to run at higher resolution can help reduce the differences at higher resolution. We will also add a recommendation in the discussion to suggest comparisons with the coarse-resolution results, which are dynamically adjusted by the amount of training data. For example, an emulator trained on ~1% of the data can be compared with a coarse-resolution data with ~10 times coarser resolution (one sample every 100 cells). This will provide a more complete context to evaluate the emulators.
> > >
> > > **3. Model performance not perfect:** Thanks for the suggestion. Yes, our current training setup uses an auto-regressive manner mirroring a similar process in the physical model. Based on the comment, we have conducted a new experiment exploring the non-autoregressive way for training and inference, by modifying the Transformer to directly forecast all 40 years’ results. The non-autoregressive implementation adopts a common decoder masking strategy to remove the need of autoregression during inference (i.e., removing the reliance on the prediction from the former time step in the sequence during training and thus the inference). Compared to the autoregressive version of the Transformer model, the results showed smaller errors on the variables that tend to grow larger over time (e.g. height RMSE from 2.888 to 2.105, AGB from 1.154 to 0.638, soil carbon from 1.710 to 0.614), and larger errors on the variables with strong seasonality and more fluctuations (e.g., GPP RMSE from 0.217 to 0.286, NPP from 0.107 to 0.143). For DeepED, several designs are more based on the autoregressive nature and would require more changes to develop a non-autoregressive version. The results suggested that future directions may consider different strategies for different types of variables to further improve the performance. We will include the results and the discussion in the paper.
> > >
> > > **4. Future simulation data:** Thanks for the suggestion. We agree that additional clarification will help people use the dataset in the intended way and we will highlight this in the paper. Meanwhile, we will append additional runs with CMIP as an addition to the current dataset so users know these are for the future forecasting and can use it to test out-of-distribution performances.
> > >
> > > **5. DeepED:** Yes, that is correct. The original DeepED is an algorithm development paper at a local region, while this work focuses on the global-scale datasets with more comparisons under more conditions.
> > >
> > > **6. Forecasting:** Thanks for the suggestion. We will then emphasize the word “emulation” in the paper instead of “carbon forecasting” to avoid the confusion.

---

> > > > ### Author Response · Authors · 2025-08-08
> > > >
> > > > Dear Reviewer,
> > > >
> > > > As today is the final day for rebuttal and discussion, we would greatly appreciate it if you could take a look at our responses and kindly let us know if more information is needed. Thanks in advance for your time and help!
> > > >
> > > > Best regards,
> > > >
> > > > The Authors

---

### Official Review · Reviewer_VBFA · 2025-07-06

**Rating:** 5
**Confidence:** 4

**Summary:**

SUMMARY: This paper offers a novel, global-scale machine learning dataset for forest carbon dynamics forecasting called Carbon-Bench. The motivation for this dataset is to reduce forecasting time (1-1.5 hour inference) by orders of magnitude compared to traditional models, such as NASA's Carbon Monitoring System used and computationally expensive (~812 CPU days for this dataset) ED model. Benchmarking revealed that among the 8 ML models tested, DeepED (specialized forest emulator) performed best on structural variables (e.g., height, biomass, soil carbon), while
transformer-based models were the best for seasonal variables (GPP, NPP, respiration).

**Additional Feedback:**

Really well-written and large effort

**Dataset Code Accessibility:**

Yes

**Ethical Considerations:**

No, there are no or only very minor ethics concerns

**Final Justification:**

After reading the authors rebuttal, which was satisfactory, and reading the other reviewers' comments and discussions on them, I believe that the mark I originally gave is suitable.

**Limitations Weaknesses:**

CONS:
* The paper though obviously of exceptional quality is at times challenging to read due to the technical expertise required (though this is acknowledged in the Dicussion)
* I am no expert in this domain, but I find a bit challenging that the entire dataset relies on outputs from a single model (i.e., ED v3.0). Does this create a potential bias?
* 0.5° resolution (~50km) is quite coarse for local-scale management decisions, but this is still understandable given the global dataset scale
* Due to computational constraints, the results are presented without error bars or statistical significance testing

**Strengths Contributions:**

PROS:
* This is a truly global dataset at 0.5° spatial resolution with 54,152 land locations
* It is also a temporally large dataset covering 40-year (1981-2021)
* It offers 100+ variables from multiple sources including meteorological data, soil properties, and CO2 concentrations
* It also offers 7 key theory-driven outputs, i.e, carbon variables (vegetation height, biomass, soil carbon, LAI, GPP, NPP, and heterotrophic respiration) generated using the validated Ecosystem Demography (ED) model. Could there be any comparison with other carbon cycle models (CLM, JULES, etc.) if it is relevant, or not?
* the data is pre-processed data with 812,280 time-series sequences, evaluation scenarios, and problem-specific metrics to be ML-ready

---

> ### Author Rebuttal · Authors · 2025-07-31
>
> We appreciate the reviewer for providing constructive comments and pointing out where we can improve it. We attempt to address your specific comments regarding limitations and questions below.
>
> **Accessibility for broader audiences**: Thank you for taking the time to read and review the paper so thoughtfully. We plan to add illustrative figures to demonstrate the framework in the appendix or dataset website.
>
> **Use of Ecosystem Demography (ED) model**: Thanks for the question. Yes the current dataset focuses on a single model, ED, as we find it is already challenging for current ML models to approximate the process. Our aim with this initial release is to establish the first global-scale ML-ready dataset for ecosystem carbon forecasting. As ED has been widely validated and adopted in both research and applications (e.g., Global Carbon Budget, NASA Global Ecosystem Dynamics Investigation (GEDI) mission, NASA Carbon Monitoring System, forest inventory of the State of Maryland), it will be valuable if newer developments in ML models can provide emulations of the process with improved accuracy. Meanwhile, we will mention the use of a single model as a current limitation and discuss future expansions with more models or model versions.
>
> **0.5° resolution**: Thanks for acknowledging the applicability of 0.5° resolution at the global scale, and we also agree that this resolution is coarse for local decision-making. In our prior work, we have conducted higher-resolution study with a much smaller scope in Maryland, US (i.e. 1km resolution [1]). Future local applications can potentially leverage models pretrained on our dataset and finetuned for finer-scales. Considering the data size, in the future we plan to include additional finer-scale data in sub-regions to support related model development.
>
> **Suggested error bars**: Thanks for the suggestion. We have added additional statistics in the Appendix based on three independent runs for selected models and scenarios.
>
>
> Reference:
>
> [1] Hurtt, G., Zhao, M., Sahajpal, R., Armstrong, A., Birdsey, R., Campbell, E., ... & Tang, H. (2019). Beyond MRV: high-resolution forest carbon modeling for climate mitigation planning over Maryland, USA. Environmental Research Letters, 14(4), 045013.

---

> > ### Comment · Reviewer_VBFA · 2025-08-03
> >
> > I would like to thank the authors for the responses, which are satisfactory, and I will keep my mark as is.

---

### Comment · Area_Chair_ZC29 · 2025-08-01
**Reviewer response to rebuttals**

Reviewers, can you please take a look at the author's rebuttal and respond as soon as possible.

---

### Note · Authors · 2025-08-16

Dear AC and Reviewers:

Thanks so much for your time and feedback on our submission. We summarize below our final remarks on the main discussion points:

**Evaluation against observations**: To prepare this dataset, we have extensively evaluated ED data in our previous works against observations, independent datasets, and atmospheric inversions (Ma et al., 2022; Ma et al., 2023; Hurtt et al., 2019; citations in detailed responses). Following the suggestions, we have additionally evaluated emulator outputs against 35 site-level observations during the rebuttal period. The results show close emulator agreement with small prediction error  (e.g., 0.1180 RMSE and 0.0578 MAE between Transformer emulator and in-situ data, and 0.1122 RMSE and 0.0505 MAE between our ED dataset and in-situ data). Fig. 1 in the paper visualizes an example comparing ED and in-situ observations at one site. The additional results will be added to the paper, and the observations used will be shared as well as part of the dataset for community use.

**0.5° resolution**: In Earth Science, 0.5°-2° datasets (ours is 0.5°) are widely used for global studies for ecosystems (e.g., Ahlström et al., 2015 in Science; Humphrey et al., 2021 in Nature). Some of the most impactful benchmark datasets (e.g. WeatherBench with 1.4° to 5.6°) also have similar or lower resolutions. This resolution is also more practical for deep learning model development, as higher resolutions become computationally challenging for GPU and I/O (e.g., as suggested by WeatherBench). We will clarify this context in the paper.

**Relation to previous work of DeepED**: The DeepED paper focuses on model development in a local northeastern U.S. region, and does not provide any public dataset. Our dataset is at the global scale with broader scenarios. DeepED model is included as one candidate approach in the evaluation. We will clarify this in the paper.

**Lower-resolution baselines**: Following the reviewer's suggestions, we added the coarser ED (2°x2°) as a baseline which led to about 16x reduction in computation cost but major error increases (e.g., 152% for height; 313% for AGB). This baseline helps better show the need for emulators. We will add this in the paper.

More details are available in the responses.

We appreciate the constructive feedback. Thanks again for your suggestions.

Best regards,

The Authors

---

### Decision · Program_Chairs · 2025-09-18

**Decision:**

Accept (poster)

**Comment:**

The majority of reviewers recommend acceptance, citing that this is a "novel and timely benchmark dataset" that captures an important and as-yet unsolved challenge for the ML community (carbon forecasting) and that the paper is well-written and clear. The AC agrees, and recommends acceptance to D&B. In particular, the discussion and rebuttal period resulted in highly constructive feedback from many of the reviewers that, when addressed and incorporated into the paper, will significantly improve the work (a win for the ML review process). There is one reviewer that disagrees, but their concerns are mainly related to the resolution proposed (.5 degrees) being insufficient to be valuable, but the authors correctly emphasize that this is well within and in fact on the high-resolution end of much of the scientific work published in this space.

===== FINAL UPDATE FROM DB Track PCs ====

The final decision for this paper has been taken by the program chairs after consultation with the SACs. All Senior Area Chairs have ranked papers according to the feedback from the AC during the review process. We decided to leave the original meta-review to reflect the opinion of the AC in light of the initial discussions with reviewers and SAC.